# pTNAS: Progressive Neural Architecture Search for Tabular Data

**Naili Xing** [1]   **Shaofeng Cai** [1]   **Lingze Zeng** [1]   **Jiaqi Zhu** [1]   **Peng LU** [2]   **Jian Pei** [3]   **Beng Chin Ooi** [2]

## Abstract

Recent advances have shifted the paradigm of tabular learning toward tabular foundation models, yet their accuracy relies on a heavy inference cost that scales poorly with context size. Deep neural networks remain a highly competitive and more efficient modeling paradigm when equipped with well-designed architectures; however, identifying such architectures in a data-adaptive and budget-aware manner remains challenging. We propose pTNAS, the first progressive neural architecture search (NAS) approach tailored for tabular data, which enables fast identification of a viable architecture and continuously improves its search performance as more budget becomes available. pTNAS adopts a filter-and-refine optimization strategy that combines efficient training-free and effective training-based architecture evaluation. In the filtering phase, we introduce pTProxy, a novel zero-cost proxy specifically designed for tabular networks that jointly captures architectural trainability and expressivity, enabling fast filtering of large architecture search spaces. In the refinement phase, pTNAS employs a fixed-budget scheduling algorithm to accurately identify the best-performing architecture from a small set of promising candidates. We further propose a budget-aware coordinator to optimize budget allocation holistically. Experiments show that pTNAS reduces the time to reach the globally best architecture by up to $82.75\times$ compared with other NAS approaches, achieves the best average predictive rank, and improves end-to-end efficiency by up to $4.78\times$ compared with TabPFN.

[1]National University of Singapore, Singapore [2]Zhejiang University, China [3]Duke University, USA. Correspondence to: Shaofeng Cai <shaofeng@comp.nus.edu.sg>.

*Proceedings of the $43^{rd}$ International Conference on Machine Learning*, Seoul, South Korea. PMLR 306, 2026. Copyright 2026 by the author(s).

## 1. Introduction

Tabular data remains the most common format in real-world applications, such as financial modeling, medical diagnosis, and recommendation systems (Cheng et al., 2024; 2025). In these high-stakes domains, even marginal gains in predictive accuracy can translate into significant economic or social value (Cai et al., 2021; Zhu et al., 2023). However, achieving peak performance on tabular data is challenging and typically requires sophisticated modeling (McElfresh et al., 2023).

Recent advances have rapidly reshaped tabular learning, shifting the dominant paradigm from robust tree ensembles (e.g., GBDTs) (Ke et al., 2017; Prokhorenkova et al., 2018) to deep tabular models and, most recently, Tabular Foundation Models (TFMs) (van Breugel & van der Schaar, 2024; Robertson et al., 2025). TFMs leverage large-scale pretraining and in-context learning (ICL) to make predictions conditioned on a small labeled sample set provided as context, reducing or even eliminating per-dataset training (Hollmann et al., 2023; Qu et al., 2025; Zhu et al., 2025). However, TFMs shift much of the computational cost to inference, exposing a critical efficiency–accuracy trade-off: while providing more labeled samples as context has the potential to improve adaptation to the target data distribution, inference latency increases with context length and can quickly become a bottleneck (Bonet et al., 2024; Mueller et al., 2025). For example, TabPFN (Hollmann et al., 2023) infers by conditioning on the context set via a pretrained prior over tabular tasks. Yet, it is typically confined to small datasets (e.g., fewer than 1,000 training samples, 100 features, and 10 classes). It remains about $10\times$ slower than comparable tree-based methods, limiting its practical use to only a few thousand training examples (Bonet et al., 2024).

The recent momentum around TFMs may suggest that tabular prediction is heading toward foundation-scale inference by default. Yet, given their inference-time cost, it is worth asking whether foundation-scale inference is truly necessary for strong tabular performance. Recent studies suggest a promising alternative: even relatively lightweight Deep Neural Networks (DNNs) (e.g., MLPs and ResNets) can be highly competitive if their architectures are properly configured (Grinsztajn et al., 2025; Gorishniy et al., 2021; Bonet et al., 2024; Mueller et al., 2025). Despite their simplicity,

their stacked nonlinear transformations can capture complex and high-order feature interactions across heterogeneous feature types (Cheng et al., 2024), while remaining much more inference-efficient than TFMs. This leads us to a natural question: *Is it possible to efficiently discover and train a data-specific DNN architecture that matches the accuracy of long-context TFMs while maintaining the efficiency of GBDTs*? Ideally, such a process would be progressive and budget-aware: it would identify a viable architecture within a short time budget and continuously improve its performance as more budget becomes available.

Recently, Neural Architecture Search (NAS) has offered a principled approach to automate DNN architecture design and shows promise in other modalities such as vision and language (Ji et al., 2025; Kang, 2025; Wang et al., 2020b). However, applying NAS to tabular prediction while delivering progressive improvements faces fundamental challenges: First, existing tabular NAS methods largely rely on training-based architecture evaluations, requiring the training of a large number of candidate architectures over many iterations (Yang et al., 2022). This makes the search process computationally expensive and undermines the progressive property in practice. Second, training-free evaluation via zero-cost proxies has shown promise in vision tasks by efficiently estimating architecture performance from initialization statistics without full training (Ji et al., 2025; Kang, 2025). Yet, to the best of our knowledge, its effectiveness on tabular data remains unexplored, and no tabular-specific zero-cost proxy has been designed. This gap is non-trivial: unlike images or texts, which possess inherent spatial or sequential structures, tabular data is characterized by heterogeneous features and a lack of structural priors, making it far from trivial to capture complex, non-intuitive feature interactions through simple architectural statistics at initialization (Borisov et al., 2024; Kang, 2025). Further, in vision tasks, zero-cost proxies are often used in relatively static roles (e.g., to pre-train or warm-start search strategies) (Mellor et al., 2021; Abdelfattah et al., 2021), when directly transferred to tabular NAS, their inherent estimation noise makes such static usage insufficient to deliver reliable progressive improvements. Additionally, the absence of a NAS benchmark dataset dedicated to tabular data further hinders systematic and reproducible research.

To this end, we present **pTNAS**, a *progressive NAS approach tailored for tabular data*. pTNAS follows a filter-and-refine optimization strategy: it first performs a coarse-grained filtering phase to efficiently explore a large set of candidate architectures and shortlist promising architectures using a tabular-specific zero-cost proxy pTProxy, and then enters a fine-grained refinement phase to accurately identify the best-performing model among the shortlisted candidates based on more expensive training-based architecture evaluation. A budget-aware coordinator holistically allocates budget across the two phases, enabling pTNAS to continuously improve search quality as more budget becomes available. In summary, this paper makes the following contributions:

- We introduce *NAS-Bench-Tabular*, the first benchmark dataset for tabular NAS, featuring over 160K unique architectures evaluated across multiple real-world datasets to facilitate reproducible research.

- Based on *NAS-Bench-Tabular*, we analyze and benchmark existing state-of-the-art zero-cost proxies from vision tasks on tabular datasets, and introduce the first tabular-specific zero-cost proxy, pTProxy, which captures both expressivity and trainability of architectures.

- We propose pTNAS, a progressive NAS approach for tabular data that introduces a novel filter-and-refine optimization strategy combining the benefits of training-free and training-based architecture evaluation.

- Extensive evaluations demonstrate that pTNAS achieves up to an $82.75\times$ speedup in search efficiency and outperforms recently proposed deep tabular models and TFMs across 8 benchmark datasets.

pTNAS is the core algorithm behind the model selection operator in NeurDB (Ooi et al., 2024; Zhao et al., 2025; Xing et al., 2024).

The rest of the paper is organized as follows: Section 2 introduces notations, Section 3 details the methodology, Section 4 presents the experiments, and Section 5 concludes the paper. Related work is presented in Appendix B.

## 2. Notation and Terminology

A typical NAS approach consists of three key components: a search space, a search strategy, and an architecture evaluation mechanism (White et al., 2023; Ren et al., 2021).

**Search Space** $\mathcal{A} = \{a\}$ is a collection of architectures, where each $a$ has a unique topology. Typically, $\mathcal{A}$ is characterized by multiple design dimensions, including depth (the number of layers), width (the number of channels or hidden units per layer), and macro-topology (the connectivity pattern between layers) (Ying et al., 2019; Siems et al., 2020). For a DNN-style search space with a fixed macro-topology, each $a$ is parameterized by the widths of $L$ hidden layers, where each width is chosen from the set of layer sizes $\mathcal{H}$. The search space therefore contains $|\mathcal{H}|^L$ architectures. For example, a search space $\mathcal{A}$ with $L = 6$ and $\mathcal{H} = 8, 16, 32, 64, 128$ contains $5^6 = 15{,}625$ architectures.

**Search Strategy** is responsible for proposing a candidate architecture $a_{i+1}$ for evaluation from the search space, denoted as $a_{i+1} = f_s(\mathcal{A}, \mathcal{S}_i)$, where $\mathcal{S}_i$ represents the state of

the search strategy at the $i$-th iteration. Its objective is to efficiently explore the search space by evaluating promising architectures (Bohdal et al., 2023; Cai et al., 2020; Yang et al., 2022). Popular search strategies include random sampling (Bergstra & Bengio, 2012), reinforcement learning (Zoph & Le, 2017), evolutionary algorithms (Ying et al., 2019; Real et al., 2019), and Bayesian optimization with HyperBand (Falkner et al., 2018).

**Architecture Evaluation** refers to assessing the performance of a given architecture. Training-based methods evaluate architectures after full training (Zoph & Le, 2017), while training-free evaluation estimates performance by computing architecture statistics at initialization from a small data batch (Tanaka et al., 2020; Ji et al., 2025; Kang, 2025). Given $a$ with parameters $\boldsymbol{\theta}$ and a batch $X_B$ of $B$ samples, a zero-cost proxy computes a score $s_a = \rho(a, \boldsymbol{\theta}, X_B)$, where $\rho(\cdot)$ is the assessment function.

## 3. Methodology

pTNAS is structured into two phases: the *coarse-grained filtering phase* and the *fine-grained refinement phase*, based on training-free and training-based architecture evaluation, respectively, and optimized holistically via a budget-aware coordinator to support progressive NAS. In the coarse-grained filtering phase, pTNAS efficiently explores the search space, directed by a search strategy using our new zero-cost proxy. Next, in the fine-grained refinement phase, pTNAS evaluates the most promising architectures accurately via training-based evaluation. A coordinator is also introduced to guide the two phases, ensuring pTNAS delivers a high-performing architecture within the time budget $T_{\max}$.

To ensure fair and consistent benchmarking across different NAS approaches, we first establish *NAS-Bench-Tabular* (Sec. 3.1). Next, we analyze zero-cost proxies in terms of trainability and expressivity, and develop a new training-free proxy for tabular architectures (Sec. 3.2). We then use this proxy for coarse-grained architecture filtering in the coarse-grained filtering phase (Sec. 3.3) and employ a scheduling algorithm to optimize the fine-grained refinement phase (Sec. 3.4). Finally, we introduce a budget-aware coordinator to facilitate progressive NAS (Sec. 3.5).

### 3.1. NAS-Bench-Tabular Design

We build *NAS-Bench-Tabular* as a tabular NAS benchmark dataset by (i) defining a discrete search space that contains diverse possible architectures, and (ii) precomputing the fully-trained performance of each architecture on multiple tabular datasets, enabling NAS approaches to directly query per-dataset architecture performance without training.

Existing studies show that lightweight DNNs with modest depth and properly tuned hidden sizes could already achieve

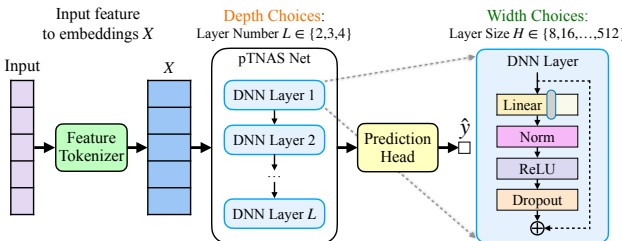

*Figure 1.* Search Space in pTNAS.

top performance on tabular data (Holzmüller et al., 2024; Yang et al., 2022; Kadra et al., 2021), and recent work (Grinsztajn et al., 2025; Mueller et al., 2025) further shows that even TFMs can be distilled into compact MLPs that preserve most of the accuracy while achieving much lower latency. Therefore, we employ a DNN backbone to construct the tabular search space, parameterized by $L$ hidden layers and a candidate set of layer sizes $\mathcal{H}$, as shown in Figure 1. Each DNN layer consists of a linear transformation, batch normalization, ReLU activation, and dropout as described in Equation 1. The search goal is then to determine the layer number and the size of each layer.

$$
\begin{aligned}
\text{pTNAS}(X) &= \text{Pred}\left(h_L(\cdots(h_1(X)))\right) \\
h_\ell(x) &= \text{Dropout}\left(\text{ReLU}\left(\text{Norm}\left(\text{Linear}(x)\right)\right)\right) \\
\text{Pred}(x) &= \text{Linear}\left(\text{ReLU}\left(\text{Norm}(x)\right)\right)
\end{aligned}
\tag{1}
$$

We adopt three widely benchmarked tabular datasets: Frappe, Diabetes, and Criteo (Yang et al., 2022). These datasets are well-known and commonly used tabular datasets (Cai et al., 2021; Luo et al., 2023) covering diverse domains (e.g., app recommendation, healthcare) with substantially different scales (about 100K to 46M samples) and feature space (369 to 5,382 features). More details, analysis, and discussions of *NAS-Bench-Tabular* are provided in Appendix D. We also include a more heterogeneous space composed of MLP, attention, transformer, and residual blocks, each with its own width. Please refer to Appendix D.4 for details.

### 3.2. pTProxy: A Tabular-Specific Zero-Cost Proxy

Theoretically, zero-cost proxies characterize two key properties of the architecture that are related to its performance: *trainability* (Wang & Fu, 2023; Shin & Karniadakis, 2020; Chen et al., 2021) and *expressivity* (Hornik et al., 1989; Wang et al., 2023b; Raghu et al., 2017).

**Trainability** measures an architecture's optimization efficiency via gradient descent, where its state at initialization often determines final performance. Since individual DNN parameters vary in their contribution to task learning, characterizing trainability requires aggregating these parameter-wise importance scores. Specifically, *synaptic*

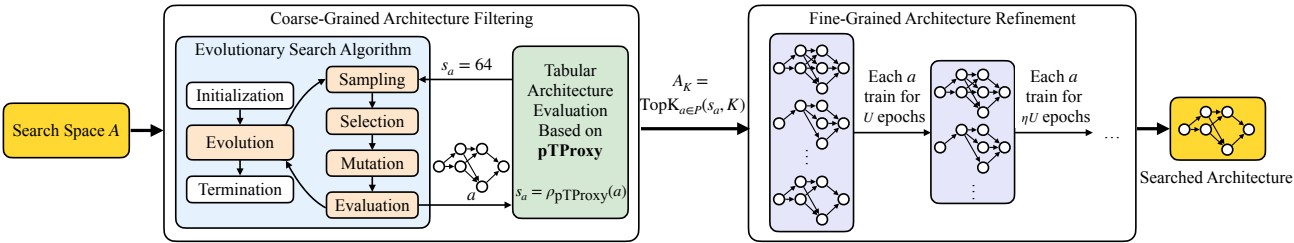

*Figure 2.* The Overview of pTNAS.

*saliency* (Tanaka et al., 2020; Lee et al., 2019) defines the importance of each parameter as $\Phi(\boldsymbol{\theta}) = f(\frac{\partial \mathcal{L}}{\partial \boldsymbol{\theta}}) \odot g(\boldsymbol{\theta})$, where $\mathcal{L}$ is the loss function and $\odot$ denotes the Hadamard product. Consequently, the trainability of an architecture can be characterized by the global aggregation of these scores (Tanaka et al., 2020; Mellor et al., 2021):

$$s_{\text{trainability}} \propto \sum_{\boldsymbol{\theta}_i} \Phi(\boldsymbol{\theta}_i) = \sum_{\boldsymbol{\theta}_i} f(\frac{\partial \mathcal{L}}{\partial \boldsymbol{\theta}_i}) \odot g(\boldsymbol{\theta}_i) \quad (2)$$

**Expressivity** characterizes the hypothesis space that an architecture can realize, specifically, the complexity of the input-to-output mappings it is capable of approximating (Zhang et al., 2021). For tabular DNNs structured as sequences of stacked nonlinear transformations, expressivity is fundamentally governed by architectural depth and width (Hanin & Rolnick, 2019). Increased depth facilitates the hierarchical composition of successive nonlinearities, while greater width expands the latent dimensionality at each transformation step. Consequently, architectures with a larger number of layers $L$ and expanded hidden dimensions within the candidate set $\mathcal{H}$ can represent more complex nonlinear functions (Zhang et al., 2021; Hanin & Rolnick, 2019). Formally, we define the expressivity as:

$$s_{\text{expressivity}} = \Psi(\{h_l\}_{l=1}^L), \quad h_l \in \mathcal{H} \quad (3)$$

where $L$ denotes the network depth, $\mathcal{H}$ represents the set of hidden layer sizes, and $\Psi$ models the inter-layer interactions, capturing how the sequence of transformations collectively defines the DNN's representation.

**pTProxy Design.** To characterize architecture performance in terms of both trainability and expressivity, we propose pTProxy for efficient tabular DNNs evaluation. pTProxy is based on *neuron saliency*, a more effective characterization of architecture performance. For the $n$-th neuron in the DNN, we quantify its saliency in the architecture, denoted as $\nu_n$. Specifically, this is computed as the product of the absolute value of the derivative of $\mathcal{L}$ with respect to the activated output of the neuron $z_n$, and the value of $z_n$ itself, i.e., $\nu_n = |\frac{\partial \mathcal{L}}{\partial z_n}| \odot z_n$, where $z_n = \sigma(\mathbf{w}\mathbf{x} + b)$, and $\mathbf{w}$ represents the incoming weights of the neuron, $\mathbf{x}$ is the neuron inputs, $b$ is the bias, and $\sigma$ is the activation function. For the ReLU activation function, $\nu_n = |\frac{\partial \mathcal{L}}{\partial z_n}| \odot z_n$ if $z_n > 0$, otherwise $\nu_n = 0$.

The raw $\nu_n$ are not naturally comparable across different depths and widths. For depth, a direct summation of $\nu_n$ across layers is inherently biased toward deeper representations. Prior work shows that small perturbations introduced at an earlier layer can be amplified by subsequent transformations, and this amplification grows with the remaining depth of the network (Raghu et al., 2017). This rapid amplification causes $\nu_n$ in later layers to appear larger, thereby dominating the aggregated score and masking contributions from earlier neurons. To mitigate such bias, we first estimate a layer-wise amplification factor via finite differences:

$$d_l = \frac{\left\| z^{(l)}(x + \epsilon \delta) - z^{(l)}(x) \right\|_2}{\epsilon} \quad (4)$$

where $z^{(l)}(x)$ is the activation vector of layer $l$, $\delta$ is sampled from a standard Gaussian distribution with the same shape as $x$, and the perturbed input is $x + \epsilon\delta$. $d_l$ denotes layer sensitivity, which typically scales with depth (Raghu et al., 2017). Then, we recalibrate the neuron saliency $\nu_n$ at layer $l$ by the inverse factor $1/d_l$. This normalization prevents late-layer saliency from dominating the aggregated score, thereby preserving early-layer contributions. For width, a layer's influence reflects not only the saliency of individual neurons but also the dimensionality of the feature space it spans. Following (Zhang et al., 2021), we scale the recalibrated $\nu_n$ by the layer width $\mathcal{K}_l$. Intuitively, given equal per-neuron saliency, a wider layer provides a larger representational subspace and thus a greater cumulative impact. Multiplying by $\mathcal{K}_l$ ensures this dimensional bandwidth is captured in the final aggregated score.

Altogether, the recalibration weight for $\nu_n$ at layer $l$ is $\frac{\mathcal{K}_l}{d_l}$. We finally perform a weighted aggregation of neuron saliency to derive a proxy score that depicts the performance of an architecture $a$ on a batch of data $X_B$, as follows:

$$
\begin{aligned}
s_a &= \rho_{\text{pTProxy}}(a; X_B) \\
&= \sum_{i=1}^B \sum_{n=1}^N \frac{\mathcal{K}_{l(n)}}{d_{l(n)}} \nu_{in} \\
&= \sum_{i=1}^B \sum_{l=1}^L \frac{\mathcal{K}_l}{d_l} \left( \sum_{n=1}^{N_l} \left| \frac{\partial \mathcal{L}}{\partial z_{in}} \right| \odot z_{in} \right)
\end{aligned} \quad (5)
$$

where $\nu_{in}$ and $z_{in}$ are the neuron saliency and activated

**Algorithm 1** The Overall Workflow of pTNAS

**Input** : Search space $\mathcal{A}$; budget $T_{\max}$; mini-batch $X_B$;
SH params $(U, \eta)$.

**Output** : Best-performing architecture $a^\star$.

/* Budget allocation based on Eq. 6 */
Choose $(M, K)$ s.t. $T_1(M) + T_2(K, U) \leq T_{\max}$

**Coarse-Grained Architecture Filtering**
  $\mathcal{P} \leftarrow$ InitPopulation$(\mathcal{A}, \rho_{\text{pTProxy}}(\cdot; X_B))$
  /* Evolutionary search guided by pTProxy (Eq. 5) */
  **while** $|\mathcal{P}| < M$ **do**
    $\mathcal{S} \leftarrow$ SampleSubset$(\mathcal{P})$
    $a \leftarrow \arg\max_{a \in \mathcal{S}} s_a$
    $a' \leftarrow$ Mutate$(a)$
    $s_{a'} \leftarrow \rho_{\text{pTProxy}}(a'; X_B)$
    $\mathcal{P} \leftarrow$ UpdatePopulation$(\mathcal{P}, (a', s_{a'}))$
  $\mathcal{A}_K \leftarrow$ TopK$(\mathcal{P}, s_a, K)$

**Fine-Grained Architecture Refinement**
  $U_{\text{cur}} \leftarrow U$;
  $\mathcal{C} \leftarrow \mathcal{A}_K$
  /* Budget scheduling based on SH */
  **while** $|\mathcal{C}| > 1$ **do**
    **foreach** $a \in \mathcal{C}$ **do**
      Train$(a, U_{\text{cur}})$
      $p_a \leftarrow \mathcal{V}(a)$
    $\mathcal{C} \leftarrow$ TopK$(\mathcal{C}, p_a, \lceil |\mathcal{C}|/\eta \rceil)$
    $U_{\text{cur}} \leftarrow U_{\text{cur}} \cdot \eta$
  $a^\star \leftarrow$ the remaining architecture in $\mathcal{C}$
**return** $a^\star$

---

output of the $n$-th neuron computed on the $i$-th sample, respectively, $N$ denotes the total number of neurons in architecture, and $N_l$ is the number of neurons in the $l$-th layer.

**Tabular-Specific Design.** pTProxy is specifically designed for tabular data. Unlike vision tasks that rely heavily on spatial priors such as translation invariance, tabular learning requires modeling implicit and high-order interactions among heterogeneous features (Song et al., 2019; Xie et al., 2021). Neurons serve as the basic feature-extraction units for modeling such interactions in tabular DNNs (Levin et al., 2023; Cai et al., 2021), combining signals from multiple input features and acting as local nonlinear detectors of useful feature patterns. Therefore, pTProxy operates at the neuron level rather than the parameter level used by many existing zero-cost proxies. Concretely, pTProxy captures both trainability and expressivity through three mechanisms. (1) Neuron saliency considers the activation value of neurons. By computing the activated neuron output $z_n$, neuron saliency captures complex and non-intuitive relationships among input features in tabular data. (2) Neuron saliency also accounts for neuron-level derivatives. A larger gradient of the loss with respect to the activation $z_n$, i.e., $\frac{\partial \mathcal{L}}{\partial z_n}$,

indicates higher importance of the features extracted by this neuron, and therefore greater significance for the prediction task. The absolute value in $\nu_n$ prevents positive and negative neuron-saliency contributions from canceling each other, ensuring that important neuron contributions are retained in the aggregated score. (3) Neuron saliency's recalibration weight, i.e., $\frac{\mathcal{K}_l}{d_l}$, is determined by the depth and width of its layer. Neurons in earlier and wider layers receive higher saliency values, highlighting their larger influence on architecture performance. More in-depth theoretical analysis of pTProxy regarding its trainability and expressivity is provided in Appendix E.

### 3.3. Coarse-Grained Architecture Filtering

While pTProxy enables fast architecture evaluation (within seconds), exhaustively scoring the entire search space $\mathcal{A}$ is computationally prohibitive. To address this, we adopt an evolutionary search algorithm (EA) (Ying et al., 2019; Real et al., 2019) to explore the discrete search space efficiently. EA maintains a population of architectures and iteratively samples a parent subset, selects the highest-scoring architecture based on pTProxy ($s_a = \rho_{\text{pTProxy}}(a; X_B)$), and applies a mutation to generate a new candidate, where each mutation randomly selects one layer and replaces its width with another candidate value from the predefined width set. For the BlockMixed space in Appendix D.4, each mutation randomly inserts or deletes a block, or changes one block attribute, including block type, width, normalization, activation, dropout, connectivity, or the auxiliary parameter of attention/transformer blocks. Each new candidate is evaluated using pTProxy and added to the evaluated candidate set. This evolutionary process continues until the target number of architectures ($M$) has been explored, as shown in Algorithm 1. Here, $M$ is determined by the coordinator (Section 3.5). The procedure then yields a ranked set of evaluated candidates. The top-$K$ architectures among these candidates are then passed to the fine-grained refinement for training-based evaluation.

### 3.4. Fine-Grained Architecture Refinement

To mitigate the estimation gap introduced by the training-free proxy pTProxy, pTNAS performs a fine-grained training of the top-$K$ architectures $\mathcal{A}_K$. To reduce the computational cost of fully training all $K$ candidates, we adopt a budget-aware scheduling strategy based on Successive Halving (SH) (Jamieson & Talwalkar, 2016). SH begins by assigning a small, equal budget to all candidates. In each round, the lower-performing fraction is discarded, while the remaining architectures receive increased training budgets. This process continues until a single best-performing architecture is identified. By progressively allocating budget, SH enables early elimination of weak candidates and concentrates more budget on high-potential candidates.

## 3.5. Progressive Neural Architecture Search

As illustrated in Figure 2, pTNAS operates in a filter-and-refine optimization strategy: a coarse-grained filtering phase for efficient exploration over the search space $\mathcal{A}$, and a fine-grained refinement phase for effective exploitation of top-$K$ promising candidates. A budget-aware coordinator is introduced to holistically allocate resources across the two phases, to maximize architecture performance under a total time budget $T_{\max}$, as detailed in Algorithm 1.

To formalize this optimization, we quantify the time of each phase. Let $t_1$ and $t_2$ denote the time required to evaluate an architecture using pTProxy and to train for one epoch, respectively. Exploring $M$ architectures in the coarse-grained filtering phase incurs a total time of $T_1 = Mt_1$.

In the fine-grained refinement phase, SH progressively narrows the top-$K$ candidate set by retaining only the top $1/\eta$ fraction after each round. Initially, each architecture is trained for $U$ epochs using a total budget of $KUt_2$. At each round, the training budget per architecture is increased, and the number of candidates is reduced by a factor of $\eta$, ensuring that every round consumes the same total budget. Given approximately $\lceil \log_\eta K \rceil$ rounds, the total time usage is $T_2 \approx KUt_2 \lceil \log_\eta K \rceil$.

The overall optimization can therefore be formally summarized as the following constrained maximization problem:

$$\max_{\Omega} \quad p = \mathcal{V}(\mathcal{S}_{\text{refinement}} \circ \mathcal{S}_{\text{filtering}}(\mathcal{A}; \Omega))$$
$$\text{s.t.} \quad T_1(M) + T_2(K, U) \leq T_{\max}$$
$$\text{where} \quad T_1 = M\,t_1, T_2 \approx K\,U\,t_2 \lceil \log_\eta K \rceil \quad (6)$$

where $\Omega = (M, K)$, $\mathcal{S}_{\text{filtering}}$ and $\mathcal{S}_{\text{refinement}}$ are selection operators that progressively map the search space $\mathcal{A}$ to the final architecture, while $\mathcal{V}$ is the function that measures the validation performance of the selected architecture.

In practice, $\eta$ controls how aggressively architectures are pruned in each round. We use the default setting $\eta = 3$, following the original Hyperband paper (Li et al., 2017). To balance the filtering and refinement phases, we further study the sensitivity of $M/K$ and $U$ to the final architecture performance, and set them empirically based on this analysis (Appendix F.3). The coordinator therefore determines $M$ and $K$ for any predefined $T_{\max}$.

## 4. Experiments

### 4.1. Effectiveness of pTProxy

We first benchmark nine vision-based zero-cost proxies against pTNAS based on *NAS-Bench-Tabular*, and quantify their ranking quality by the *Spearman Rank Correlation Coefficient (SRCC)* between proxy scores and the ground-truth AUC. A robust zero-cost proxy exhibits consistently

high correlations across diverse data distributions.

As shown in Table 1, while NASWOT, SNIP, and SynFlow consistently achieve SRCCs above 0.6, pTProxy outperforms all others with an average rank of 1.0. Besides, pTProxy maintains small SRCC variance across datasets, which confirms its robustness and transferability in characterizing tabular architecture performance. The superior performance of pTProxy stems from its ability to simultaneously characterize both trainability and expressivity, as theoretically established in Section 3.2 and Appendix E.

### 4.2. Progressive NAS on Tabular Data

We then benchmark pTNAS against two other NAS methods proposed for tabular data, i.e., TabNAS (Yang et al., 2022) and EA-NAS (a training-based baseline using evolutionary search (Ying et al., 2019)). By varying the time budget $T_{\max}$ from seconds to hours, we investigate: (i) Efficiency: How much time does each NAS approach require to identify architectures approaching the global optimum? (ii) Progressive property: Can the pTNAS discover comparably better-performing architectures within a given budget, and does performance improve steadily as more budget becomes available?

The results are shown in Figure 3. In terms of efficiency, pTNAS significantly outperforms EA-NAS, achieving speedups of $82.75\times$, $1.75\times$, and $69.44\times$ on Frappe, Diabetes, and Criteo, while reaching strong AUCs of 0.9814, 0.6750, and 0.8033, respectively. This advantage stems from avoiding the heavy overhead of training-based evaluations, which hinders both EA-NAS and TabNAS. Instead, pTNAS holistically optimizes training-free and training-based evaluations, enabling rapid exploration over large architecture search spaces and effective pruning of suboptimal candidate architectures.

Regarding progressive property, pTNAS consistently discovers better-performing architectures than EA-NAS and TabNAS under various $T_{\max}$ and exhibits a stable upward trajectory as $T_{\max}$ increases. Both EA-NAS and TabNAS often require 5–10 minutes to evaluate a single architecture, while pTNAS provides a viable architecture even under a small $T_{\max}$. This capability is enabled by the budget-aware coordinator (Section 3.5), which dynamically partitions $T_{\max}$ between exploration and exploitation. By combining fast proxy-based filtering via pTProxy with more precise refinement, pTNAS maintains strong performance across all $T_{\max}$.

### 4.3. Comparison with Existing Tabular Models

Moreover, we benchmark pTNAS against existing NAS methods and four representative categories of tabular models: Classical Models (CM), including LR (Cox, 1958),

*Table 1.* **SRCC summary of zero-cost proxies across benchmark datasets.** For each proxy, we report the mean and standard deviation of SRCC computed over these datasets. We also report the average rank, where methods are ranked *per dataset* by descending |SRCC| (higher is better), and the ranks are then averaged across datasets.

| | GradNorm | NASWOT | NTKCond | NTKTrace | NTKTrAppx | Fisher | GraSP | SNIP | SynFlow | **pTProxy** |
|---|---|---|---|---|---|---|---|---|---|---|
| Mean SRCC | 0.40 | 0.65 | -0.66 | 0.47 | 0.15 | 0.38 | -0.24 | 0.70 | 0.74 | 0.82 |
| Std. SRCC | 0.05 | 0.05 | 0.11 | 0.08 | 0.16 | 0.14 | 0.04 | 0.09 | 0.05 | 0.08 |
| Avg. Rank | 7.3 | 4.0 | 4.3 | 6.3 | 9.3 | 8.0 | 9.0 | 3.3 | 2.3 | 1.0 |

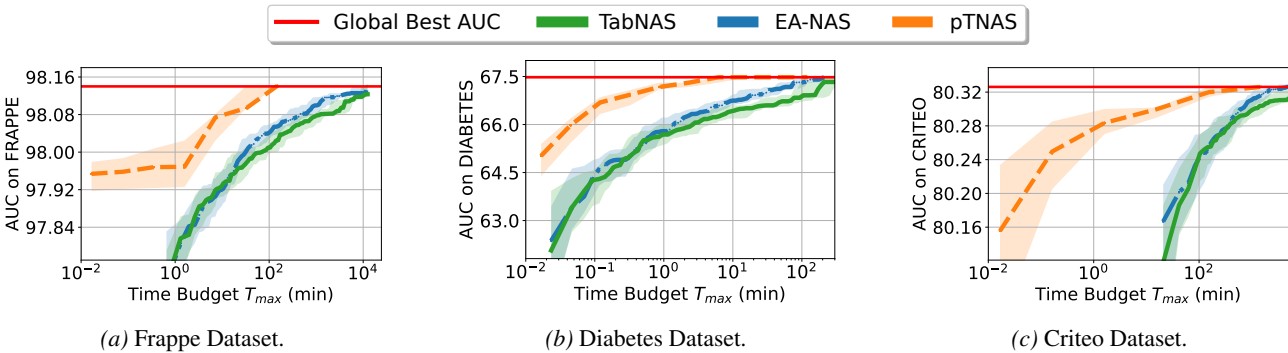

*Figure 3.* Progressive search performance of pTNAS compared with EA-NAS and TabNAS.

RF (Breiman, 2001), CatBoost (Prokhorenkova et al., 2018), and LightGBM (Ke et al., 2017); Tabular Foundation Models (TFM), including TabPFN (Hollmann et al., 2023) and TabICL (Qu et al., 2025); Deep Tabular Models (DTM), including DNN (LeCun et al., 2015), DeepFM (Guo et al., 2017), FTTrans (Gorishniy et al., 2021), and ARM-Net (Cai et al., 2021); and Large Language Models (LLM), including TP-BERTa (Yan et al., 2024), Nomic (Nussbaum et al., 2025), and BGE (Xiao et al., 2024). For all NAS methods, including pTNAS, we set the search budget to 10s based on the observed tuning and training cost of LightGBM in our comparison (10s in total for classification tasks), thereby ensuring a fair comparison under a matched computational budget. We evaluate their predictive performance on eight datasets covering both classification and regression tasks. Detailed descriptions of these datasets are deferred to Appendix C.1.

### 4.3.1. EFFECTIVENESS

Table 2 summarizes the results. Among CMs, advanced tree-based methods such as CatBoost and LightGBM clearly outperform linear models (LR) and random forests (RF), indicating that non-linear feature interactions and robust handling of categorical variables are critical.

TFMs, including TabPFN and TabICL, perform strongly on classification tasks. TabPFN achieves an average normalized score close to the best results, while TabICL ranks among the top despite being limited to classification. However, their performance in regression is less stable and suf-

fers from task-type restrictions, e.g., TabICL is inherently incompatible with regression tasks.

DTMs such as DNN, DeepFM, FT-Transformer, and ARM-Net demonstrate competitive performance across both task types. However, these models rely on fixed backbone architectures, which may not generalize optimally across diverse tabular distributions.

LLM-based approaches such as TP-BERTa, Nomic, and BGE treat tabular rows as serialized text and struggle with high-cardinality fields and input-length constraints. Thus, they underperform across both classification and regression.

TabNAS and EA-NAS perform poorly because the short budget (10s) allows them to explore only a limited number of architectures. pTNAS achieves the best overall performance in Table 2, ranking first on average across both classification and regression. This suggests that searching for a data-specific DNN architecture can be an effective way to improve performance across heterogeneous tabular distributions. We also visualize the per-dataset performance on all tabular datasets in Figure 4. Clearly, pTNAS consistently ranks among the top-performing methods and remains very close to the best method even when it is not ranked first.

### 4.3.2. EFFICIENCY

The efficiency comparison of pTNAS is shown in Figure 5. We evaluate end-to-end efficiency across three stages: neural architecture search, fitting (i.e., context preprocessing for TFM and full training for others), and inference.

*Table 2.* **Effectiveness evaluation of pTNAS against existing methods on eight datasets.** For each dataset, methods are first ranked by their raw MAE/AUC values. We report the average rank within each task type, and overall across all available datasets for each method, where a lower rank indicates better performance. We also report an average normalized score to reflect how close a method is to the best result on each dataset: for regression, the score is best MAE/MAE; for classification, it is $(AUC - 0.5)/(best AUC - 0.5)$, where a higher value is better. The two statistics are complementary: average rank reflects relative ordering, while the normalized score reflects the magnitude of the gap to the best method.

| Task/Metric | | Regression Task | | Classification Task | | Overall | |
|---|---|---|---|---|---|---|---|
| Type | Method | Avg. Rank (↓) | Avg. Norm. Score (↑) | Avg. Rank (↓) | Avg. Norm. Score (↑) | Avg. Rank (↓) | Avg. Norm. Score (↑) |
| CM | LR (Cox, 1958) | 12.25 | 0.7959 | 11.25 | 0.8694 | 11.75 | 0.8326 |
| | RF (Breiman, 2001) | 10.63 | 0.8338 | 12.75 | 0.8485 | 11.69 | 0.8411 |
| | CatBoost (Prokhorenkova et al., 2018) | 5.50 | 0.9246 | 7.00 | 0.9105 | 6.25 | 0.9176 |
| | LightGBM (Ke et al., 2017) | 5.63 | **0.9578** | 7.00 | 0.9125 | 6.31 | 0.9351 |
| TFM | TabPFN (Hollmann et al., 2023) | 9.88 | 0.8576 | 5.00 | **0.9486** | 7.44 | 0.9031 |
| | TabICL* (Qu et al., 2025) | - | - | **4.50** | 0.9439 | **4.50** | **0.9439** |
| DTM | DNN (LeCun et al., 2015) | 5.25 | 0.9343 | 8.25 | 0.8909 | 6.75 | 0.9126 |
| | DeepFM (Guo et al., 2017) | 7.50 | 0.9214 | 10.50 | 0.8581 | 9.00 | 0.8898 |
| | FTTrans (Gorishniy et al., 2021) | **4.13** | 0.9471 | 8.00 | 0.8936 | 6.06 | 0.9204 |
| | ARM-Net (Cai et al., 2021) | 5.50 | 0.9450 | 6.75 | 0.9229 | 6.13 | 0.9339 |
| LLM | TP-BERTa (Yan et al., 2024) | 13.75 | 0.6296 | 16.00 | 0.0922 | 14.71 | 0.3993 |
| | Nomic (Nussbaum et al., 2025) | 12.25 | 0.6535 | 13.75 | 0.6941 | 13.00 | 0.6738 |
| | BGE (Xiao et al., 2024) | 10.50 | 0.7865 | 13.25 | 0.7819 | 11.88 | 0.7842 |
| NAS | TabNAS (Yang et al., 2022) ($T_{max} = 10s$) | 6.50 | 0.9370 | 6.00 | 0.9262 | 6.25 | 0.9316 |
| | EA-NAS ($T_{max} = 10s$) | 9.00 | 0.8488 | 5.00 | 0.9322 | 7.00 | 0.8905 |
| | **pTNAS** ($T_{max} = 10s$) | **1.75** | **0.9891** | **1.00** | **1.0000** | **1.38** | **0.9946** |

∗ TabICL is inherently limited to classification tasks and does not apply to regression-based tasks.

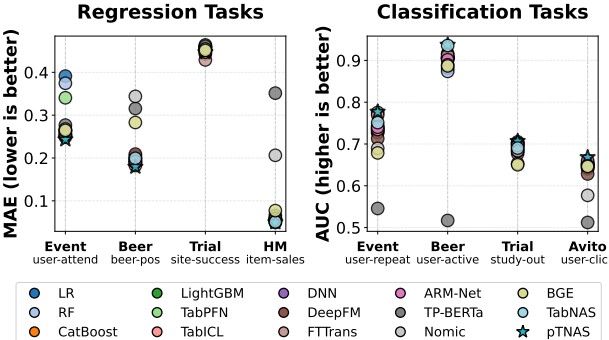

*Figure 4.* **Per-dataset performance of all eight tabular datasets.** Each point denotes one tabular model on one dataset. In general, pTNAS achieves competitive performance across datasets.

The results reveal a clear trade-off between per-task adaptability and end-to-end latency. CMs, such as CatBoost and LightGBM, offer fast inference due to their tree-based design, but exhibit highly variable training efficiency. For instance, CatBoost encounters a major bottleneck on the Trial (site-success) dataset, even with early stopping. This inefficiency arises from recursive split-finding over complex relational structures and high-cardinality categorical features, which do not scale well during training.

TFMs, such as TabPFN, are theoretically zero-shot, but introduce substantial overhead in both the fit and inference

stages. The fitting stage requires encoding the entire training set into a transformer context, leading to significant latency as the dataset size increases. More critically, TabPFN's inference is consistently the slowest among all baselines due to expensive transformer forward passes that scale quadratically with sequence length.

In contrast, pTNAS implements an efficient NAS-based method that completes NAS within 10s and produces dataset-specialized DNNs. For example, on the Event *user-attendance* dataset with $T_{max} = 10s$, the coordinator spends less than 1s determining the time allocation based on Equation 6, allocates $M = 484$ for filtering ($T_1 = 1.6s$), and keeps $K = 8$ architectures for refinement ($T_2 = 7.2s$). While introducing a small search overhead, it maintains moderate fitting time and achieves inference speed comparable to optimized CM baselines.

Overall, pTNAS delivers a superior efficiency for tabular analytics. All these results support our central claim: **effectively discovering dataset-specific DNN architectures is crucial for realizing competitive tabular performance without incurring foundation-scale inference costs.**

### 4.4. Extended Analyses and Results

We provide additional analyses to support the experimental results in the main text, with full details deferred to the ap-

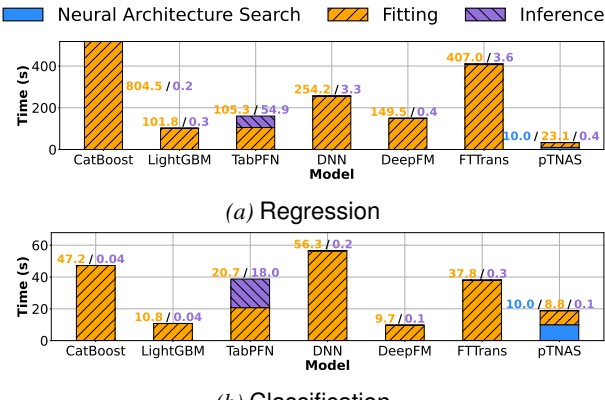

*(a)* Regression

*(b)* Classification

*Figure 5.* **Efficiency evaluation of pTNAS against existing tabular models on eight datasets.** We select representative tabular models from each model group based on their global ranking to ensure competitive baselines. For both regression and classification tasks, we report the average runtime across all datasets in the corresponding task, decomposed into search/selection (NAS), fitting, and inference. Fitting corresponds to context preprocessing for TFMs and full training for other models. Overall, pTNAS (pT-NAS) is **4.78×** and **12.25×** faster than TabPFN and FTTrans on regression, and **2.05×** and **2.02×** faster on classification, respectively (based on total time). Full results are reported in Table 17.

pendix. We begin by reporting supplementary benchmark construction and search-space details of *NAS-Bench-Tabular* in Appendix D. We then present ablations for evaluating pT-Proxy in Appendix F.1, including parameter positivity, initialization methods, batch size, and the recalibration weight used for aggregating neuron saliency. In addition, we compare pTProxy with other zero-cost proxies through correlation visualizations in Appendix F.6.

Next, we provide extended analysis of the filter-and-refine optimization strategy and the budget-aware coordinator, including sensitivity to $(M, K, U)$ and coordinator variants, in Appendix F.2 and Appendix F.3. We further include additional baseline comparisons in Appendix F.5 (e.g., different combinations of training-free and training-based NAS strategies, and one-shot NAS via weight sharing). Finally, the full per-dataset effectiveness results and the corresponding efficiency measurements used to compute the aggregated statistics are reported in Appendix G.

## 5. Conclusion

In this work, we introduce a progressive NAS approach pTNAS tailored for tabular data. pTNAS equips machine learning practitioners with the capability of ascertaining high-performance architectures within any given time budget and further refining these architectures as larger time budgets are given. We first design a search space, denoted as *NAS-Bench-Tabular*, to serve as a benchmarking platform for diverse NAS algorithms on tabular data. Then,

we conduct an evaluation of several zero-cost proxies on tabular data and propose a tabular-specific zero-cost proxy, pTProxy, which characterizes both the trainability and expressivity of an architecture. Based on these foundations, we present pTNAS, which leverages the advantages of efficient training-free and effective training-based architecture evaluation through a novel filter-and-refine optimization strategy with holistic optimization. Empirical results demonstrate that pTNAS substantially accelerates the search for high-performing architectures compared with existing tabular NAS methods. Moreover, pTNAS outperforms recently proposed deep tabular models across multiple benchmark datasets. Despite these strengths, pTNAS currently focuses on search efficiency and predictive performance, while explicitly incorporating deployment constraints such as memory footprint remains an important direction for future work.

## Acknowledgements

We dedicate this work to the memory of Professor Beng Chin Ooi, whose wisdom, generosity, and vision have profoundly shaped our research journey. His guidance and legacy will continue to inspire us.

We thank Zhaojing Luo for his early contributions.

This research is supported by the National Research Foundation, Singapore and Infocomm Media Development Authority under its Trust Tech Funding Initiative, and the National Natural Science Foundation of China (624B2027). Any opinions, findings and conclusions or recommendations expressed in this material are those of the author(s) and do not reflect the views of the National Research Foundation, Singapore and Infocomm Media Development Authority.

## Impact Statement

Tabular data is a core component of many real-world information systems and is commonly stored and managed in relational database management systems (RDBMSs). Improving learning methods for tabular data can therefore affect a broad range of applications, including business analytics, healthcare operations, and financial services, where structured records are prevalent. This work contributes a method for automating tabular model design with an emphasis on computational efficiency. By reducing the need for extensive manual tuning and expensive trial-and-error search, the proposed approach can lower the barrier to obtaining competitive models under limited compute budgets.

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

# Appendix

# A. Notations

*Table 3.* Summary of Notation and Terminology.

| | |
|---|---|
| $T_{\max}$ | Time budget |
| $\mathcal{A} = \{a\}$ | Search space and architecture |
| $L$ | Number of layers in the DNN |
| $h_l$ | the $l$-th layer in the DNN |
| $\mathcal{H}$ | A candidate set of layer sizes |
| $|\mathcal{H}|^L$ | Number of candidate architectures |
| $f_s$ | Search strategy |
| $\mathcal{S}_i$ | State of the search strategy at the $i$-th iteration. |
| $\boldsymbol{\theta}$ | Parameter of an architecture |
| $B, X_B$ | Batch size. A batch of data samples |
| $N$ | Number of neurons of the whole architecture |
| $N_l$ | Number of neurons of the $l$-th layer of the architecture |
| $\mathcal{L}$ | Loss function |
| $s_a$ | zero-cost proxy score of architecture |
| $\rho(\cdot)$ | zero-cost proxy assessment function |
| $\odot$ | Hadamard product |
| $\Theta$ | NTK metrics |
| $\Phi$ | Synaptic saliency |
| $\nu_n$ | Neuron saliency of the $n$-th neuron computed on a batch of samples |
| $\nu_{in}$ | Neuron saliency of the $n$-th neuron computed on the $i$-th sample |
| $z_n$ | Activated output of the $n$-th neuron computed on a batch of samples |
| $z_{in}$ | Activated output of the $n$-th neuron computed on the $i$-th sample |
| $\mathbf{w}$ | Incoming weights of the neuron |
| $\sigma$ | Activation function |
| $\mathbf{x}$ | Neuron inputs |
| $\mathcal{K}_l$ | Number of neurons in the $l$-th layer |
| $z^{(l)}(x)$ | Activated representation (activation vector) at layer $l$ given input $x$ |
| $\delta$ | Standard Gaussian perturbation with the same shape as $x$ |
| $\epsilon$ | Perturbation magnitude (a small positive scalar) |
| $d_l$ | Layer-wise amplification/sensitivity estimated by finite difference |
| $M$ | Number of architectures explored in the coarse-grained filtering phase |
| $K$ | Number of architectures exploited in the fine-grained refinement phase |
| $U$ | Computational unit in the fine-grained refinement phase |
| $t_1$ | Time to score an architecture |
| $t_2$ | Time to train an architecture for a single epoch |
| $T_1$ | Time allocated to the coarse-grained filtering phase |
| $T_2$ | Time allocated to the fine-grained refinement phase |
| $\Omega$ | Joint search configuration tuple, defined as $\{M, K, U\}$ |
| $H$ | Hessian vector |
| $\mathcal{N}$ | Number of architecture encodings |
| $\eta$ | $\frac{K}{\eta}$ Architectures to keep per-round |

# B. Literature Review

### B.1. DNN on Tabular Data

Different approaches have attempted to apply DNN techniques to tabular data, ranging from DNN design (Levin et al., 2023; Gorishniy et al., 2021; Luo et al., 2021; Zhu et al., 2026), Automated Machine Learning (AutoML) on tabular data (Fusi et al., 2018; Olson & Moore, 2016; Yang et al., 2019), to Tabular Foundation Models (Qu et al., 2025; Hollmann et al.,

2023; Robertson et al., 2025). TFMs leverage large-scale pretraining and in-context learning (ICL) to make predictions conditioned on a small labeled sample set provided as context, reducing or even eliminating per-dataset training. However, TFMs push much of the computational cost to inference, exposing a critical efficiency–accuracy trade-off: while providing more labeled samples as context has the potential to improve adaptation to the target data distribution, inference latency increases with context length and can quickly become a bottleneck (Bonet et al., 2024; Mueller et al., 2025).

In contrast, dataset-specific DNNs can provide fast inference after fitting, but their performance is highly dependent on the choice of architecture and training configuration. This motivates automated architecture discovery for tabular data, where the goal is to identify compact and high-performing DNNs for each dataset under practical search budgets.

### B.2. Neural Architecture Search (NAS)

Neural Architecture Search (NAS) is designed to automate the discovery of architectures optimized for specified datasets, thereby eliminating the need for manual design and experimentation. The process often entails the search for connection patterns within a predefined architectural backbone, such as Fully-connected neural network (Yang et al., 2022) or cell-based convolutional neural network (Ying et al., 2019; Dong & Yang, 2020).

A critical component of NAS is the architecture evaluation. Initial research in NAS (Zoph & Le, 2017; Baker et al., 2017) primarily relied on the time-consuming and resource-intensive process of fully training each architecture to convergence. Several methods have been proposed to address this issue and to mitigate costs (Lee et al., 2023; Shala et al., 2023; Li et al., 2023; Bohdal et al., 2023). They generally fall into three categories: NAS with performance prediction, NAS based on weight sharing, and NAS with training-free architecture evaluation.

Performance prediction in NAS involves training a model to forecast the final performance of an architecture based on features derived from hyperparameters, architectural structures, and partially trained architectures. This category of methods shows improvements in searching efficiency but it is limited in tuning the predictors and is hard to enhance generalizability (Akhauri & Abdelfattah, 2024; Kadlecová et al., 2024).

In contrast, NAS based on weight sharing seeks to identify a subgraph within a larger computation graph (Pham et al., 2018; Casarin et al., 2025). This allows multiple sampled subgraphs sharing the same computation unit to utilize a common set of weights. However, the individual sampling and training procedure of each discrete subgraph leads to an increased number of architectures to be trained. And inheriting weights from the larger computational graph does not necessarily ensure improved training efficiency (Chu et al., 2021). To further reduce the number of architectures requiring training (Liu et al., 2019) proposed shifting the search from a discrete space to a continuous one. This enables gradient optimization to expedite the search process.

Last, zero-cost proxies estimate architecture performance by calculating certain statistics of the architecture at initialization without requiring full training (Ji et al., 2025; Li et al., 2023; Kang, 2025; Oh et al., 2025). One of the main advantages of the training-free evaluation is its extremely high computational efficiency, requiring only a single forward and/or backward computation.

However, NAS on tabular data has been relatively less explored. Recently, AgEBO-Tabular (Égelé et al., 2021) and TabNAS (Yang et al., 2022) investigated the application of NAS on tabular data to achieve more efficient and higher-performing architectures. Specifically, AgEBO-Tabular integrates NAS with aging evolution in a search space that includes multiple branches and hyperparameter tuning using Bayesian optimization. In comparison, TabNAS aims to identify high-performing architectures from Fully-connected neural network under specified resource constraints by utilizing reinforcement learning with reject sampling. It shows that simple multiple layers Fully-connected neural network can already yield outstanding performance. However, both approaches rely on training-based architecture evaluation and cannot achieve progressive NAS on tabular data. Although progressive NAS has been studied in other domains, such as PNAS (Liu et al., 2018), these methods are not directly applicable to tabular data: (1) they are often designed for vision-specific, cell-based CNN search spaces rather than tabular DNN architectures; (2) their search procedures remain training-based, which is inefficient under the tight budgets required by practical tabular analytics; and (3) their predictors typically rely on architecture encodings rather than tabular-specific, data-aware signals, and therefore do not explicitly capture properties central to tabular learning.

These limitations motivate progressive tabular NAS that can use efficient architecture filtering while retaining training-based refinement only for promising candidates.

### B.3. Architecture Properties

Architecture performance is influenced by two factors: trainability and expressivity. Trainability measures the extent to which gradient descent can effectively optimize the architecture. Expressivity denotes the complexity of the function that the architecture can model. Many training-free evaluation approaches estimate the architecture's performance by characterizing both properties.

More recently, TE-NAS (Chen et al., 2021) proposes to quantify the expressivity of a ReLU-based DNN by computing the number of linear regions that the architecture can divide for a batch of data. Likewise, NASWOT (Mellor et al., 2021) characterizes expressivity by measuring the distance between the vectors of activation patterns for any two samples within a batch. A greater distance suggests a higher capability to distinguish different samples, indicating good expressivity.

The application of the Neural Tangent Kernel (NTK) as a measure of trainability has recently been explored, given a batch of data $X_B$, NTK (Jacot et al., 2018; Arora et al., 2019; Allen-Zhu et al., 2019) characterizes the complexities of the training dynamics at initialization, which is defined as

$$\Theta(X_B, X_B; \boldsymbol{\theta}) = \bigtriangledown_{\boldsymbol{\theta}} f(X_B; \boldsymbol{\theta}) \bigtriangledown_{\boldsymbol{\theta}} f(X_B; \boldsymbol{\theta})^T \tag{7}$$

NTK-related proxies are adopted in many training-free evaluation approaches, such as NTKTrace (Shu et al., 2022b), NTKTraceAppx (Shu et al., 2022a), and NTKCond (Chen et al., 2021).

The trainability has also been studied in the context of network pruning (Tanaka et al., 2020; Wang et al., 2020a; Lee et al., 2019), which identifies and prunes less significant parameters. The notion of synaptic saliency (Tanaka et al., 2020) is proposed to quantify each parameter's importance, defined as

$$\Phi(\boldsymbol{\theta}) = f(\frac{\partial \mathcal{L}}{\partial \boldsymbol{\theta}}) \bigodot g(\boldsymbol{\theta}) \tag{8}$$

Different zero-cost proxies basically differ in $f(\cdot)$ and $g(\cdot)$, for SNIP (Lee et al., 2019)

$$\Phi(\boldsymbol{\theta}) = |\frac{\partial \mathcal{L}}{\partial \boldsymbol{\theta}}| \bigodot |\boldsymbol{\theta}| \tag{9}$$

for GraSP (Wang et al., 2020a)

$$\Phi(\boldsymbol{\theta}) = -(H \frac{\mathcal{L}}{\partial \boldsymbol{\theta}}) \bigodot \boldsymbol{\theta} \tag{10}$$

and for SynFlow (Tanaka et al., 2020)

$$\Phi(\boldsymbol{\theta}) = \frac{\partial \mathcal{L}}{\partial \boldsymbol{\theta}} \bigodot \boldsymbol{\theta} \tag{11}$$

where $H$ is the Hessian vector. Similarly, Fisher (Turner et al., 2020) quantifies the performance by aggregating layer Fisher information (Theis et al., 2018).

We provide the full summarization of different zero-cost proxies in Table 4, with their evaluation metrics, complexity, score computation definition, and the characterized properties of DNN.

Taken together, pTNAS advances existing work in three aspects. First, unlike traditional tabular models with static architectures (Table 2), pTNAS searches for dataset-specific DNN architectures. Second, unlike existing tabular NAS methods such as TabNAS that rely on expensive training-based search, pTNAS introduces a training-free filtering paradigm for efficient NAS on tabular data and combines it with lightweight refinement. Third, whereas existing zero-cost proxies are primarily designed for vision-oriented settings, pTProxy is tailored to tabular data by characterizing properties central to tabular deep learning, including expressivity induced by high-order feature interactions. To the best of our knowledge, this is the first work in this direction.

## C. Experiment Setup

### C.1. Datasets and Preprocessing

We evaluate our approach on 11 datasets, including 3 widely-used tabular datasets (Frappe, UCI Diabetes, and Criteo) and 8 realistic multi-table datasets sourced from the RelBench benchmark (Fey et al., 2024). RelBench is designed for efficient

*Table 4.* A Comparison of Different Zero-Cost Proxies.

| Zero-Cost Proxy | Evaluation Metric | Complexity | Computation | Property |
|---|---|---|---|---|
| GradNorm | Frobenius norm | 1FC+1BC | $s_a = \left\| \frac{\partial \mathcal{L}}{\partial \boldsymbol{\theta}} \right\|_F$ | Express. |
| NASWOT | Hamming distance | 1FC | $s_a = \log |K_H|$ | Express. |
| NTKCond | Neural tangent kernel | 1FC+1BC | $s_a = \frac{\lambda_{\max}(\Theta)}{\lambda_{\min}(\Theta)}$ | Train. |
| NTKTrace | Neural tangent kernel | 1FC+1BC | $s_a = \|\Theta\|_{\text{trace}}$ | Train. |
| NTKTraceAppx | Neural tangent kernel | 1FC+1BC | $s_a = \|\Theta_{\text{appx}}\|_{\text{trace}}$ | Train. |
| Fisher | Hadamard product | 1FC+1BC | $s_a = \sum_{l=1}^{L} \left( \frac{\partial \mathcal{L}}{\partial ac_l} ac_l \right)^2$ | Train. |
| GraSP | Hessian vector product | 1FC+1BC | $s_a = \sum - \left( H \frac{\partial \mathcal{L}}{\partial \boldsymbol{\theta}} \right) \odot \boldsymbol{\theta}$ | Train. |
| SNIP | Hadamard product | 1FC+1BC | $s_a = \sum \left| \frac{\partial \mathcal{L}}{\partial \boldsymbol{\theta}} \odot \boldsymbol{\theta} \right|$ | Train. |
| SynFlow | Hadamard product | 1FC+1BC | $s_a = \sum \frac{\partial \mathcal{L}}{\partial \boldsymbol{\theta}} \odot \boldsymbol{\theta}$ | Train. |
| WeightNorm | Frobenius norm | 1FC | $s_a = \|\boldsymbol{\theta}\|_F$ | Express. |
| pTPProxy | Hadamard product | 2FC+1BC | $s_a = \sum_n \left| \frac{\partial \mathcal{L}}{\partial z_n} \right| \odot z_n$ | Train. & Express. |

$\mathcal{L}$: loss function. $\boldsymbol{\theta}$: architecture parameters. $\Theta$: NTK matrix of the architecture.
$\odot$: Hadamard product. $s_a$: the score of an architecture $a$.
$\lambda$: eigenvalue of the NTK matrix. $H$: Hessian vector. $L$: number of architecture layers.
$\| \cdot \|_F$: Frobenius norm. $ac_l$: activation saliency of layer $l$.
$FC$: forward computation. $BC$: backward computation.

and reproducible evaluation of end-to-end learning over relational databases, covering diverse real-world domains. Table 5 reports the key statistics of all datasets, and detailed descriptions are provided below.

For each multi-table dataset, we transform the relational database into a target-table learning setup via a DFS-style feature construction procedure (Kanter & Veeramachaneni, 2015). Starting from the target table, we traverse the schema graph and generate relational features by aggregating values from neighboring tables along join paths using predefined operators (e.g., count, sum/mean, and max/min). Following the temporal split, all features for each example are computed using only historical records available before its prediction time, and the same transformation is applied to validation/test without accessing future information. This yields a unified feature representation for downstream models while preserving informative context from the original relational structure.

**Frappe** [1] is a dataset from the real-world application recommendation scenario, which incorporates context-aware app usage logs consisting of 96,203 tuples from 957 users across 4,082 apps used in various contexts. For each positive app usage log, Frappe generates two negative tuples, resulting in a total of 288,609 tuples. The learning objective is to predict app usage based on the context, encompassing 10 semantic attribute fields with 5,382 distinct numerical and categorical embedding vectors.

**UCI Diabetes** [2] (Diabetes) encompasses a decade (1999-2008) of clinical diabetes encounters from 130 US hospitals. This dataset aims to analyze historical diabetes care to enhance patient safety and deliver personalized healthcare. With 101,766 encounters from diabetes-diagnosed patients, the primary learning objective is to predict inpatient readmissions. This dataset consists of 43 attributes and 369 distinct numerical and categorical embedding vectors, including patient demographics and illness severity factors like gender, age, race, discharge disposition, and primary diagnosis.

**Criteo** [3] is a CTR benchmark consisting of attribute values and click feedback for millions of display advertisements. The learning objective is to predict if a user will click a specific ad in the context of a webpage. This dataset has 45,840,617 tuples across 39 attribute fields with 2,086,936 distinct numerical and categorical embedding vectors. These include 13 numerical attribute fields and 26 categorical attributes.

---

[1] https://www.baltrunas.info/research-menu/frappe

[2] https://archive.ics.uci.edu/ml/datasets

[3] https://labs.criteo.com/2014/02/kaggle-display-advertising-challenge-dataset/

*Table 5.* Dataset Statistics.

| Dataset | #Class | #Feature | Task | Domain | Train | Valid | Test | Total |
|---|---|---|---|---|---|---|---|---|
| Frappe | 2 | 10 | Cls. | AppRec | 202,027 | 57,722 | 28,860 | 288,609 |
| Diabetes | 2 | 43 | Cls. | Health | 81,412 | 10,177 | 10,177 | 101,766 |
| Criteo | 2 | 39 | Cls. | CTR | 33,003,326 | 8,250,124 | 4,587,167 | 45,840,617 |
| Event (user-repeat) | 2 | 25 | Cls. | RecSys | 3,842 | 268 | 246 | 4,356 |
| Event (user-attend) | – | 19 | Regr. | RecSys | 19,239 | 2,013 | 1,958 | 23,210 |
| Beer (user-active) | 2 | 47 | Cls. | Review | 16,656 | 2,794 | 3,558 | 23,008 |
| Beer (beer-pos) | – | 43 | Regr. | Review | 45,922 | 12,858 | 7,218 | 66,998 |
| Trial (study-outcome) | 2 | 50 | Cls. | Health | 11,994 | 960 | 825 | 13,779 |
| Trial (site-success) | – | 14 | Regr. | Health | 100,000 | 19,740 | 22,617 | 142,357 |
| Avito (user-click) | 2 | 15 | Cls. | Online | 59,454 | 21,183 | 47,996 | 128,633 |
| Avito (ad-ctr) | – | 18 | Regr. | Online | 5,100 | 1,766 | 1,816 | 8,682 |
| HM (user-churn) | 2 | 13 | Cls. | Retail | 100,000 | 76,556 | 74,575 | 251,131 |
| HM (item-sales) | – | 32 | Regr. | Retail | 100,000 | 100,000 | 100,000 | 300,000 |

**Event** [4] is an event recommendation dataset derived from a mobile social networking app Hangtime, capturing users' social plans, invitations, and interactions. It connects users, events, and friendships to model evolving engagement behavior over time. The dataset spans user activity records in 2012 and supports two prediction objectives: (1) user-attendance: estimating how many events a user will respond to in the following week (regression task), and (2) user-repeat: predicting whether a user will attend another event within a week after prior participation (classification task).

**Avito** [5] is an online advertising marketplace dataset derived from the Avito advertisement platform, capturing user queries, ad metadata, locations, and interaction logs. It connects users, ads, and sessions to model evolving browsing and engagement behavior. The dataset covers user activity from late April to May 2015 and supports two prediction objectives: (1) user-clicks: predicting whether a user will click multiple ads in the next four days (classification task), and (2) ad-ctr: estimating an advertisement's click-through rate (regression task).

**Trial** [6] is a relational dataset curated from the AACT initiative that captures study protocols, participating sites, medical conditions, interventions, and outcome reports. It links studies, sites, and interventions to model trial progress and safety, covering records from 2020 to 2021. The dataset supports two prediction objectives: (1) study-outcome: predicting whether a trial achieves its primary outcome (classification task), and (2) site-success: forecasting site-level success rates over the subsequent year (regression task).

**Beer** (McAuley & Leskovec, 2013) is a relational dataset of beer reviews linking users, beers, and drinking venues (e.g., bars/breweries). It captures user ratings and textual feedback across seasons and supports two prediction objectives: (1) user-active (classification task): predicting whether a user will post more than 10 reviews in the next season, and (2) beer-positive (regression task): estimating a beer's positive-rating ratio, where ratings above 3.5 are considered positive.

**HM** [7] is a retail-market dataset consisting of customer profiles, item metadata, and fine-grained transaction records. It captures purchase behaviors at the item level and enables forecasting future sales dynamics. We consider a prediction objective defined over a one-week horizon, where the goal is to estimate the sales volume of each product in the upcoming week. The dataset supports two prediction objectives: (1) item-sales: estimating the sales volume of each product in the upcoming week, computed as the total transaction value aggregated over that week for each item (regression task), and (2) user-churn: predicting whether a customer becomes inactive in future transactions (classification task).

---

[4] https://www.kaggle.com/c/event-recommendation-engine-challenge

[5] https://www.kaggle.com/c/avito-context-ad-clicks

[6] https://aact.ctti-clinicaltrials.org/

[7] https://www.kaggle.com/competitions/h-and-m-personalized-fashion-recommendations

*Table 6.* Training Hyperparameters.

| Dataset | batch size | learning rate | learning rate schedule | optimizer | training epoch | iteration per epoch | loss function |
|---|---|---|---|---|---|---|---|
| Frappe | 512 | 0.001 | cosine decay | Adam | 20 | 200 | BCELoss |
| Diabetes | 1024 | 0.001 | cosine decay | Adam | 1 | 200 | BCELoss |
| Criteo | 1024 | 0.001 | cosine decay | Adam | 10 | 2000 | BCELoss |

### C.2. Training Protocol and Hyperparameters

For all baseline models as listed in Table 2, we adopt a unified training protocol. We use the Adam optimizer with a fixed learning rate of $10^{-3}$ (no learning-rate schedule) and a batch size of 256. Training runs for up to 200 epochs with early stopping (patience 10), and we cap the computation per epoch by using at most 20 batches to standardize per-epoch computation across methods.

For the classification task, we optimize BCEWithLogits loss and report ROC-AUC; for regression objectives, we optimize L1 loss and report MAE. Following our implementation, dropout is deactivated for the regression task to improve training stability. The final architecture is selected by validation performance and then evaluated on the held-out test split.

### C.3. Implementation Details

All experiments in this paper are implemented in PyTorch and run on a local server equipped with an Intel(R) Xeon(R) Silver 4214R CPU (12 cores), 128 GB memory, and eight NVIDIA GeForce RTX 3090 GPUs. To support reproducibility and fair comparisons in NAS research, we release the source code, reproduction scripts, and links to the precomputed benchmark/result artifacts at `https://github.com/NLGithubWP/pTNAS`. The repository contains the training and evaluation pipelines, example outputs, and step-by-step instructions in `README.md`; the same `README.md` provides download links for *NAS-Bench-Tabular* and other large artifacts.

## D. NAS-Bench-Tabular and Search Space

### D.1. Training Hyperparameters for Building *NAS-Bench-Tabular*

We build *NAS-Bench-Tabular* using three datasets: Frappe, UCI Diabetes, and Criteo. Training hyperparameters (e.g., batch size, training epochs/iterations, and learning rate) can interact with architecture size, materially affecting measured performance. To obtain reliable training configurations, for each dataset, we first select three representative architectures from our search space, covering small, medium, and large model sizes, and perform a grid search to tune training hyperparameters for each representative architecture separately.

We require that, under their tuned hyperparameters, these three representative architectures achieve DNN performance consistent with prior results on the same datasets (Cai et al., 2021; Kadra et al., 2021; Klambauer et al., 2017; Fernández-Delgado et al., 2014). The resulting three sets of hyperparameters (small/medium/large) for each dataset are summarized in Table 6. For Adam, we use $\beta_1 = 0.9$, $\beta_2 = 0.999$, decay $= 0$, and $\epsilon = 1e-8$, following (Yang et al., 2022; Kingma & Ba, 2015).

Using the above protocol, we fully train every architecture in the search space on each dataset. Each architecture is trained using the hyperparameter set corresponding to its size category (small, medium, or large), where the category is determined by bucketing architectures by their parameter counts using two fixed thresholds.

For each architecture-dataset pair, we record five metrics: training/validation AUC, training/validation loss, and training time for all trained architectures, where training AUC is used for analyzing overfitting behavior and space statistics, and all NAS evaluations query validation AUC.

### D.2. Search Space Design and Characterization

We consider a DNN-based search space that varies the hidden-layer widths while keeping other backbone components fixed. Specifically, we fix the number of hidden layers $L$ to be four for all three tabular datasets and define an architecture $a$ by its layer-width tuple $a = (h_1, h_2, h_3, h_4)$, where each $h_\ell$ is selected from a candidate set $\mathcal{H}$. We fix the activation (ReLU),

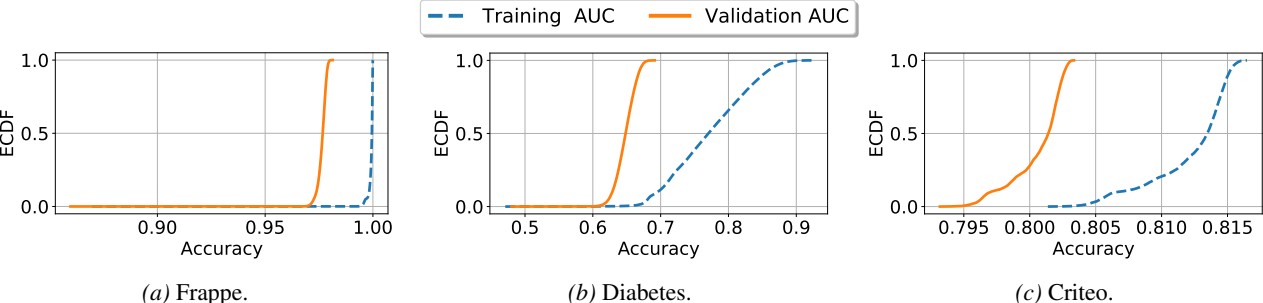

*Figure 6.* The empirical cumulative distribution function (ECDF) of the training and validation AUC recorded across all architectures in *NAS-Bench-Tabular*.

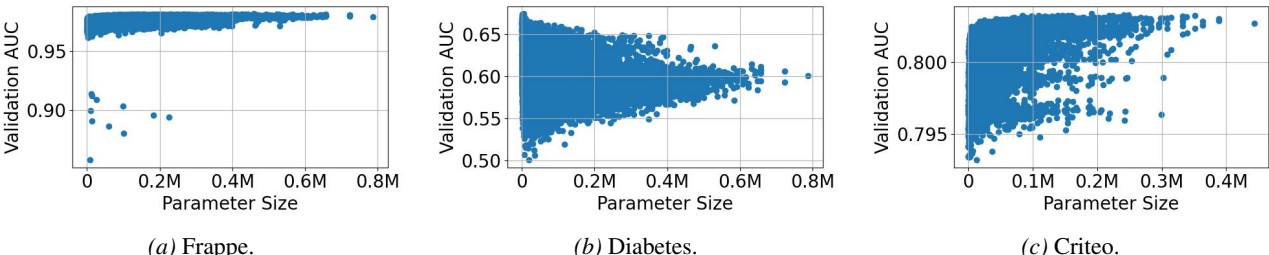

*Figure 7.* Validation AUC vs. the number of trainable parameters across all architectures in the search space.

output layer, and training-time regularization choices (e.g., dropout rate), and only vary hidden widths. We set $\mathcal{H}$ as follows:

- **Frappe and Diabetes:** $\mathcal{H} = [8, 16, 24, 32, 48, 64, 80, 96, 112, 128, 144, 160, 176, 192, 208, 224, 240, 256, 384, 512]$.

- **Criteo:** $\mathcal{H} = [8, 16, 32, 48, 64, 112, 144, 176, 240, 384]$.

Accordingly, the search space size is $|\mathcal{H}|^L$. For Criteo, the space is smaller with only $10^4 = 10,000$ architectures, while the best-performing architecture found already achieves performance comparable to prior reported results under similar settings.

We characterize the performance landscape of the *NAS-Bench-Tabular* search space from two complementary perspectives: (1) the overall distribution of achievable performance across all architectures, and (2) the relationship between model scale (parameter count) and validation performance. Figure 6 reports the empirical cumulative distribution function (ECDF) of the recorded training and validation AUC across all architectures, while Figure 7 correlates each architecture's parameter count with its validation AUC.

From Figure 6, median validation AUCs are 0.9772, 0.6269, and 0.8014 for Frappe, Diabetes, and Criteo, while optimal architectures achieve 0.9814, 0.6750, and 0.8033, respectively. These results are consistent with the performance benchmarks reported in prior work (Cai et al., 2021; Wang et al., 2023a), validating our training configurations and confirming that *NAS-Bench-Tabular* captures a meaningful spectrum of architecture qualities. Moreover, Figure 7 shows that the parameter count does not strongly correlate with validation AUC, suggesting that simply scaling the model is insufficient to reliably improve tabular performance. Together, these observations motivate topology searching (i.e., allocating widths across layers) to identify high-performing architectures and justify the necessity of NAS for tabular DNNs.

### D.3. Benchmarking Search Strategies on *NAS-Bench-Tabular*

Finally, we benchmark four representative search strategies on *NAS-Bench-Tabular*, serving as reference baselines for evaluating future NAS algorithms on our datasets. The four search strategies include Random Search (RS), Evolutionary Algorithm (EA), Reinforcement Learning (RL), and Bayesian Optimization with HyperBand (BOHB). All strategies explore the same search space defined in Appendix D.2 and use training-based evaluation by directly querying the validation AUC from *NAS-Bench-Tabular*.

For EA, we set the population size to 10 and the sample size to 3. In the RL setup, we employ a categorical distribution for

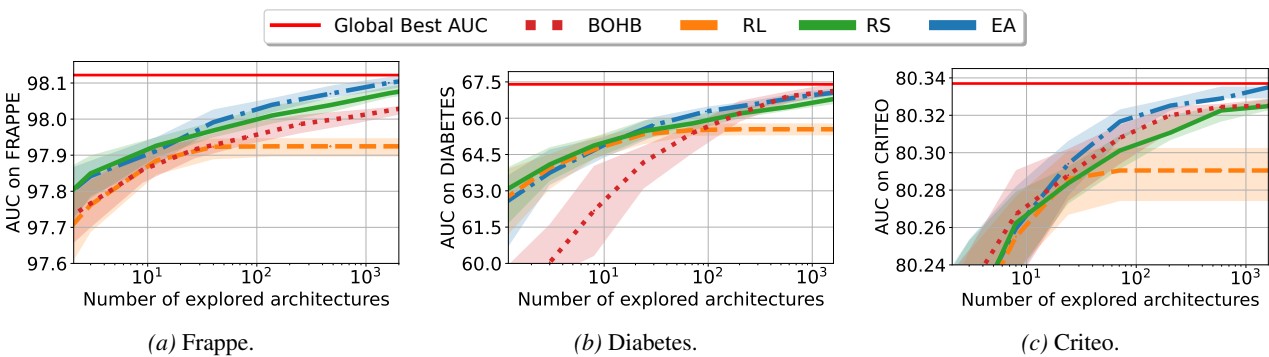

*(a)* Frappe.        *(b)* Diabetes.        *(c)* Criteo.

*Figure 8.* Benchmarking four search strategies on *NAS-Bench-Tabular*. The x-axis denotes the number of explored architectures, and the y-axis denotes the median value of the best validation AUC achieved across 100 runs.

each hidden-layer size and optimize the probabilities using policy gradient methods. For BOHB, we follow the standard setup that couples Bayesian optimization with HyperBand-style resource allocation.

Figure 8 reports the anytime performance of these strategies. The x-axis denotes the number of explored architectures, and the y-axis denotes the median of the best validation AUC achieved across 100 runs.

With *NAS-Bench-Tabular*, NAS algorithms can significantly reduce search times to seconds by querying recorded evaluations. For example, each of the four strategies can explore over 1k architectures in around 15 seconds in our implementation by querying validation AUCs, highlighting the benefit of recorded evaluations. As shown in Figure 8, EA targets high-performing architectures after exploring around $10^3$ candidates and is consistently sample-efficient across datasets. We therefore adopt the evolutionary algorithm (EA) as the search strategy in pTNAS.

### D.4. Generalization to Additional Search Spaces: ResDNN and BlockMixed

To examine whether pTProxy and pTNAS generalize beyond the fixed-depth MLP search space used by *NAS-Bench-Tabular*, we introduce two additional search spaces with richer architectural variation.

- **ResDNN** is a DNN-based residual search space with approximately 2.5K architectures. It varies network depth, hidden width, and residual skip connections, thereby extending the *NAS-Bench-Tabular* MLP space with variable-depth connectivity patterns.

- **BlockMixed** is a more heterogeneous search space with approximately 1K architectures. It is composed of MLP, attention, transformer, and residual blocks, where each block has its own width. In addition, BlockMixed varies block-level normalization (LayerNorm or BatchNorm), activation function (ReLU, GELU, or SiLU), and dropout rate (0.0, 0.1, or 0.2). Figure 9(a) illustrates the BlockMixed design.

We evaluate these two spaces on four datasets that cover different task types and data regimes. Specifically, we use Event (*user-attendance*), a relatively small-scale, high-dimensional regression task; Avito (*user-clicks*), a large-scale, low-dimensional classification task; and two additional datasets, Avito (*ad-ctr*) and H&M (*user-churn*). This setup allows us to test whether the proxy ranking quality and the downstream search behavior remain robust when both the search space and the dataset distribution change.

Table 7 compares pTProxy with the top-5 existing zero-cost proxies from Table 1. For each dataset and search space, we rank the proxies by the absolute SRCC between proxy score and final test performance, and report the average rank across the four datasets. pTProxy achieves the best average rank on both ResDNN and BlockMixed, indicating that its neuron-level saliency remains effective under variable-depth residual architectures and heterogeneous block-level architectures.

Figure 10 visualizes the search behavior of pTNAS on these additional spaces. As the number of explored architectures $M$ increases, pTNAS generally improves the best-so-far architecture quality across both ResDNN and BlockMixed, showing that the search process remains effective beyond the original *NAS-Bench-Tabular* space. For BlockMixed, examples of the architectures selected at the end of the search are shown in Figure 9(b).

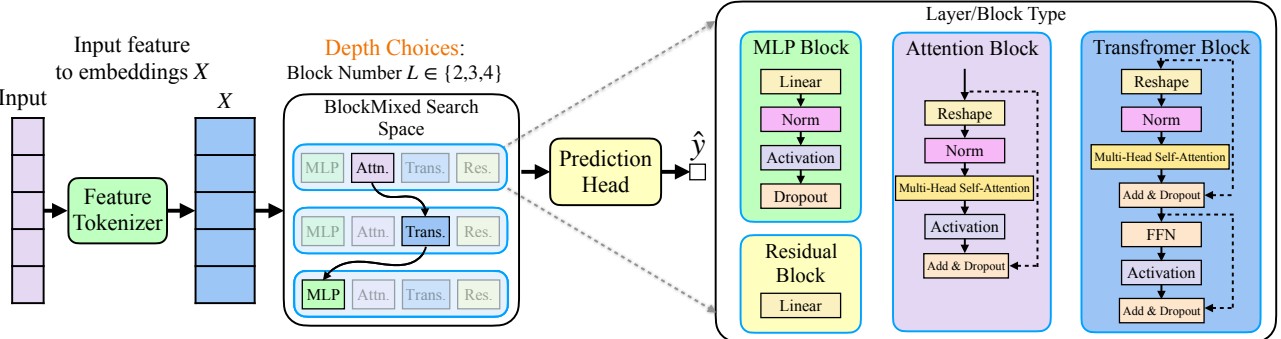

(a): **BlockMixed Search Space:** a backbone built by stacking 2 - 4 heterogeneous blocks (MLP, attention, transformer, or residual (skip)), each with its width selected from {16, 24, …, 192}.

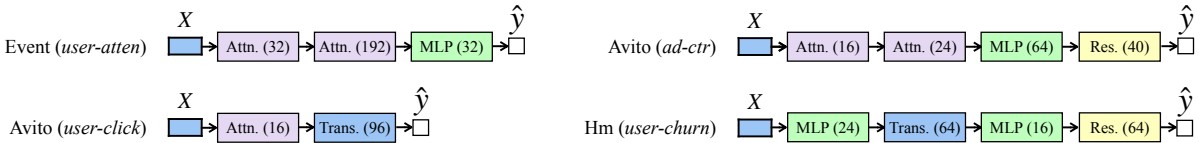

(b): **Searched Architecture for each dataset.**

*Figure 9.* BlockMixed search space and searched architectures. (a) Each architecture is constructed from heterogeneous block types, including MLP, attention, transformer, and residual blocks, with block-specific widths and training-time design choices. (b) Examples of final architectures selected by pTNAS after search.

*Table 7.* Average SRCC rank of pTProxy and the top-5 existing proxies over four datasets for each additional search space. Proxies are ranked by |SRCC| on each dataset; a lower rank indicates stronger correlation with final architecture performance.

| Search Space | SynFlow | SNIP | NTKCond | NASWOT | NTKTrace | pTProxy |
|---|---|---|---|---|---|---|
| ResDNN | 2.25 | 4.00 | 5.25 | 2.75 | 5.75 | **1.00** |
| BlockMixed | 4.00 | 3.50 | 3.00 | 5.50 | 3.75 | **1.25** |

Overall, the results across *NAS-Bench-Tabular*, ResDNN, and BlockMixed show that pTNAS is not tied to a single fixed-depth MLP search space. Instead, pTProxy continues to provide useful architecture rankings in more diverse spaces.

## E. Theoretical Analysis of pTProxy

As defined in Section 3.2, the neuron saliency of the $n$-th neuron is

$$\nu_n = \left| \frac{\partial \mathcal{L}}{\partial z_n} \right| \odot z_n, \tag{12}$$

where $z_n = \sigma(h_n)$ is the post-activation value and

$$h_n = \sum_u \mathbf{w}_{nu}^{in} z_u + b \tag{13}$$

is the pre-activation. Here $\mathbf{w}_{nu}^{in}$ denotes the incoming weight from neuron $u$ in the previous layer to neuron $n$.

We further denote $\mathbf{w}_{vn}^{out}$ as the outgoing weight from neuron $n$ to neuron $v$ in the next layer, whose pre-activation is

$$h_v = \sum_k \mathbf{w}_{vk}^{out} z_k + b. \tag{14}$$

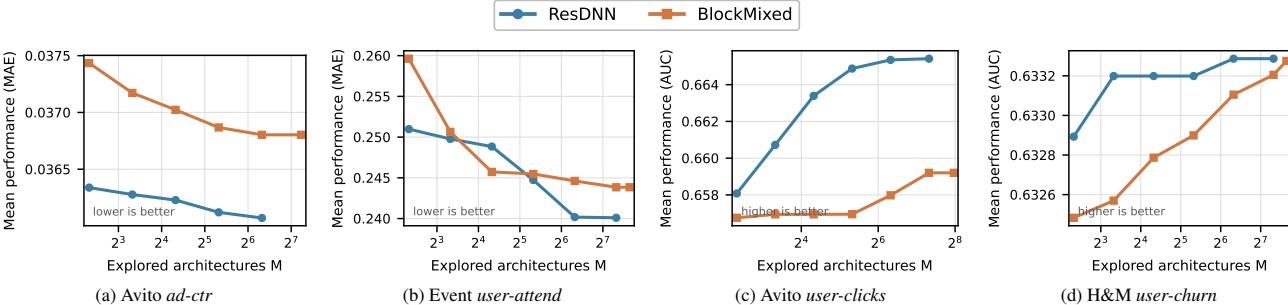

*Figure 10.* **Search performance of pTNAS on two additional search spaces.** We report the mean best-so-far performance over five random seeds as the number of explored architectures $M$ increases.

By the chain rule,

$$\frac{\partial \mathcal{L}}{\partial z_n} = \sum_v \frac{\partial \mathcal{L}}{\partial h_v} \frac{\partial h_v}{\partial z_n}. \tag{15}$$

Since $\frac{\partial h_v}{\partial z_n} = \mathbf{w}_{vn}^{out}$, we obtain

$$\frac{\partial \mathcal{L}}{\partial z_n} = \sum_v \frac{\partial \mathcal{L}}{\partial h_v} \mathbf{w}_{vn}^{out}. \tag{16}$$

Therefore, the neuron saliency can be written as

$$\nu_n = \left| \sum_v \frac{\partial \mathcal{L}}{\partial h_v} \mathbf{w}_{vn}^{out} \right| \odot z_n. \tag{17}$$

For ReLU networks, $z_n = \sigma(h_n) \geq 0$. Moreover, since $z_n$ is a scalar, the Hadamard product $\odot$ reduces to scalar multiplication, yielding

$$\nu_n = \left| \sum_v \frac{\partial \mathcal{L}}{\partial h_v} \mathbf{w}_{vn}^{out} z_n \right| = \left| \sum_v \left( \frac{\partial \mathcal{L}}{\partial h_v} z_n \right) \mathbf{w}_{vn}^{out} \right|. \tag{18}$$

Next, note that

$$\frac{\partial h_v}{\partial \mathbf{w}_{vn}^{out}} = z_n, \tag{19}$$

which implies $\frac{\partial \mathcal{L}}{\partial \mathbf{w}_{vn}^{out}} = \frac{\partial \mathcal{L}}{\partial h_v} \frac{\partial h_v}{\partial \mathbf{w}_{vn}^{out}} = \frac{\partial \mathcal{L}}{\partial h_v} z_n$. Substituting into Eq. 18, we obtain

$$\nu_n = \left| \sum_v \left( \frac{\partial \mathcal{L}}{\partial h_v} \frac{\partial h_v}{\partial \mathbf{w}_{vn}^{out}} \right) \mathbf{w}_{vn}^{out} \right| = \left| \sum_v \frac{\partial \mathcal{L}}{\partial \mathbf{w}_{vn}^{out}} \mathbf{w}_{vn}^{out} \right|. \tag{20}$$

Evidently, the term $\frac{\partial \mathcal{L}}{\partial \mathbf{w}_{vn}^{out}} \mathbf{w}_{vn}^{out}$ matches the notion of synaptic saliency used in SynFlow (Tanaka et al., 2020), which was originally introduced to quantify the importance of individual parameters. Eq. 20 shows that the neuron saliency of neuron $n$ is the absolute value of the aggregated SynFlow-style saliency over all its outgoing parameters. Compared with SynFlow, pTProxy performs aggregation at the neuron level, and the absolute value prevents sign cancellation across outgoing connections, which leads to a more consistent proxy for identifying promising DNN architectures.

Moreover, for homogeneous activations we have $\sigma(\alpha h) = \alpha \sigma(h)$ for $\alpha > 0$, and thus one can write $z_n = \sigma(h_n) = $

$\sigma'(h_n)\, h_n$ almost everywhere (e.g., for ReLU away from the kink at $h_n = 0$). For a fixed neuron $n$, this yields

$$
\begin{aligned}
\nu_n &= \left| \frac{\partial \mathcal{L}}{\partial z_n} \right| \odot z_n = \left| \frac{\partial \mathcal{L}}{\partial h_n} \right| \odot h_n = \left| \frac{\partial \mathcal{L}}{\partial h_n} h_n \right| \\
&= \left| \frac{\partial \mathcal{L}}{\partial h_n} \left( \sum_u \mathbf{w}_{nu}^{in} z_u + b \right) \right| \\
&= \left| \sum_u \left( \frac{\partial \mathcal{L}}{\partial h_n} \mathbf{w}_{nu}^{in} z_u \right) + \frac{\partial \mathcal{L}}{\partial h_n} b \right| \\
&= \left| \sum_u \left( \frac{\partial \mathcal{L}}{\partial h_n} \frac{\partial h_n}{\partial \mathbf{w}_{nu}^{in}} \mathbf{w}_{nu}^{in} \right) + \frac{\partial \mathcal{L}}{\partial h_n} b \right| \\
&= \left| \sum_u \left( \frac{\partial \mathcal{L}}{\partial \mathbf{w}_{nu}^{in}} \mathbf{w}_{nu}^{in} \right) + \frac{\partial \mathcal{L}}{\partial h_n} b \right|.
\end{aligned} \tag{21}
$$

The term $\frac{\partial \mathcal{L}}{\partial \mathbf{w}_{nu}^{in}} \mathbf{w}_{nu}^{in}$ is again a synaptic-saliency form for the incoming parameters of neuron $n$. Therefore, $\nu_n$ aggregates synaptic saliency over both outgoing (Eq. 20) and incoming (Eq. 21) connections at a neuron-wise granularity, which is well suited for capturing complex feature interactions in tabular models.

Finally, we perform the weighted aggregation of neuron saliency across neurons and samples to obtain the pTProxy score for an architecture $a$:

$$
s_a = \sum_{i=1}^{B} \sum_{n=1}^{N} \frac{\mathcal{K}_{l(n)}}{d_{l(n)}} \nu_{in} = \sum_{i=1}^{B} \sum_{n=1}^{N} \frac{\mathcal{K}_{l(n)}}{d_{l(n)}} \left| \frac{\partial \mathcal{L}}{\partial z_{in}} \right| \odot z_{in}, \tag{22}
$$

where $l(n)$ denotes the layer containing neuron $n$. This provides a training-free proxy that captures both trainability and expressivity signals at initialization.

## F. Extended Experiments

This section provides additional empirical evidence to complement the main results. We organize the experiments into four parts: (i) ablations of pTProxy, (ii) ablations of the two-phase design and the budget-aware coordinator, (iii) comparisons with additional baselines, and (iv) a broader comparison and analysis of different zero-cost proxies.

### F.1. Ablation Studies of pTProxy

In this section, we present ablation studies that justify key design choices in computing pTProxy at initialization. We focus on how implementation details (e.g., parameter positivity, initialization, batch size, and recalibration weights) affect the correlation between the pTProxy score and the true architecture performance.

**Impacts of Parameter Positivity.** As neuron saliency is computed at architecture initialization (Section 3.3), we examine the impact of the parameter sign on the effectiveness of pTProxy. Specifically, we score each architecture using either the default parameters or their absolute values, and compare the resulting correlations with architecture AUC. For consistency, we fix the batch size to $B = 32$ and use Xavier initialization. Table 8 shows that enforcing parameter positivity significantly improves correlation on all datasets, with the largest gain on Criteo. Therefore, we set parameters to their absolute values before computing pTProxy.

*Table 8.* Impacts of Parameter Positivity.

| Dataset | Frappe | Diabetes | Criteo |
|---|---|---|---|
| pTProxy with positive $\mathbf{w}$ | 0.8364 | 0.7124 | 0.8978 |
| pTProxy with default $\mathbf{w}$ | 0.5175 | 0.5901 | 0.5020 |

**Impacts of Initialization Method.** Based on the advantage of enforcing parameter positivity, we further examine how initialization affects the pTProxy correlation. With $B = 32$, we compare LeCun (LeCun et al., 2002), Xavier (Glorot &

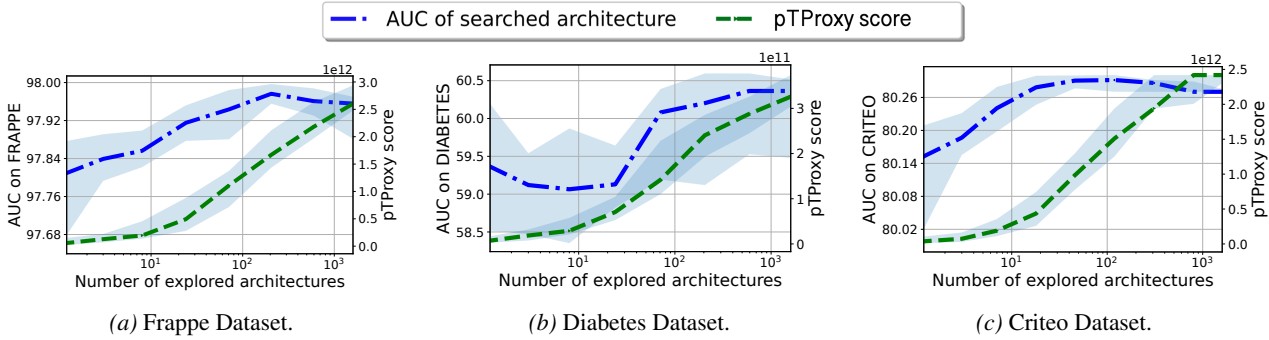

*(a)* Frappe Dataset.      *(b)* Diabetes Dataset.      *(c)* Criteo Dataset.

*Figure 11.* Relationship between the pTProxy score and validation AUC under coarse-grained filtering-only search as the explored budget increases.

Bengio, 2010), and He (He et al., 2015) initializations. Table 9 shows that He initialization consistently performs best on Criteo and Diabetes, which is expected since He is designed for ReLU networks and our search space backbone uses ReLU (Section 3.1). We therefore adopt the He initialization as the default.

*Table 9.* Impacts of Initialization method.

| Dataset | LeCun (LeCun et al., 2002) | Xavier (Glorot & Bengio, 2010) | He (He et al., 2015) |
|---|---|---|---|
| Frappe | 0.8175 | 0.8364 | 0.8150 |
| Diabetes | 0.7335 | 0.7124 | 0.7336 |
| Criteo | 0.8823 | 0.8978 | 0.9005 |

**Impacts of Batch Size $B$.** We evaluate the sensitivity of pTProxy to different batch sizes, varying $B$ from 4 to 128. For each setting, we fix the He initialization and repeat the experiment five times, reporting the median correlation. Table 10 shows that batch size has a limited influence on the correlation, suggesting that pTProxy can be computed efficiently using small batches.

*Table 10.* Impacts of Batch Size.

| Dataset | $B = 4$ | $B = 8$ | $B = 16$ | $B = 32$ | $B = 64$ | $B = 128$ |
|---|---|---|---|---|---|---|
| Frappe | 0.8154 | 0.8152 | 0.8150 | 0.8150 | 0.8150 | 0.8149 |
| Diabetes | 0.7335 | 0.7336 | 0.7335 | 0.7336 | 0.7336 | 0.7336 |
| Criteo | 0.8990 | 0.8998 | 0.9008 | 0.9005 | 0.9009 | 0.9009 |

**Impact of Recalibration Weight of Neuron Saliency.** pTProxy aggregates neuron saliency across layers with a recalibration weight $\frac{\mathcal{K}_l}{d_l}$ (Section 3.3). We compare pTProxy with two variants: (1) recalibrated only by width $\mathcal{K}$, and (2) recalibrated only by depth $\frac{1}{d_l}$. Table 11 shows that considering both width and depth consistently yields the highest correlations across datasets.

*Table 11.* Impacts of Recalibration Weights of Neuron Saliency.

| Dataset | recalibrated by $\mathcal{K}$ | recalibrated by $\frac{1}{d_l}$ | recalibrated by $\frac{\mathcal{K}_l}{d_l}$ |
|---|---|---|---|
| Frappe | 0.7007 | 0.7296 | 0.8155 |
| Diabetes | 0.6772 | 0.6900 | 0.7336 |
| Criteo | 0.6402 | 0.6414 | 0.8990 |

## F.2. Component Analysis of Two-Phase Design

**Necessity of the fine-grained refinement Phase**. To evaluate whether the coarse-grained filtering phase alone can achieve progressive NAS, we run coarse-grained filtering-only search and track the architectures it selects as the budget increases. Figure 11 shows that although the search identifies architectures with higher pTProxy scores when increasing the budget, these higher scores do not consistently translate into higher validation AUC. In particular, increasing the budget may even lead to selecting architectures with higher pTProxy scores but inferior AUC. This confirms that coarse-grained filtering-only search does not reliably satisfy the progressive requirement, and motivates the need for the fine-grained refinement phase that performs training-based evaluation (Section 3.4).

**Noise Impact on fine-grained refinement Phase**. Given that pTProxy scores are imperfect estimates, we investigate whether injecting random noise when selecting the top-$K$ candidates could improve robustness. We fix the total search time to 20 minutes and vary the *Random Noise Degree* (RND). For each RND, we keep the top $(1 - \text{RND})K$ architectures among the top-$K$ ranked by pTProxy and randomly replace $\text{RND} \cdot K$ architectures with samples from the search space, then evaluate these $K$ candidates in fine-grained refinement. Table 12 shows that introducing noise does not improve AUC, and smaller noise yields better performance. This further confirms the effectiveness of the coordinator in selecting top-$K$ candidates for fine-grained refinement.

Table 12. Impact of noise degrees on the searched AUC. Search for 20 mins.

|  | RND = 100% | RND = 70% | RND = 50% | RND = 30% | RND = 0% |
|---|---|---|---|---|---|
| AUC | 0.9792 | 0.9794 | 0.9796 | 0.9799 | **0.9802** |

## F.3. Hyperparameter Analysis of the Coordinator

**Sensitivity of $M/K$ and $U$**. As explained in Section 3.5, pTNAS employs a filter-and-refine optimization strategy with joint optimization under $T_{\max}$. The primary challenge is choosing $M$ and $K$ given $T_{\max}$, where $M$ is the number of candidates explored in the coarse-grained filtering phase using pTProxy and $K$ is the number of promising architectures exploited in the fine-grained refinement phase by training. Exploring many architectures while neglecting fine-grained refinement (e.g., $K = 1$) is efficient but can select sub-optimal architectures due to proxy noise, whereas training too many architectures (e.g., $K = M$) violates the progressive requirement.

A second challenge is the trade-off between $K$ and $U$ under a fixed training budget for fine-grained refinement, where $U$ denotes the computation unit (epochs) used to evaluate each architecture in fine-grained refinement. Training each architecture longer improves evaluation fidelity but reduces the number of explored candidates; training shorter enables exploring more candidates but with noisier estimates.

Therefore, we first examine the trade-off between $K$ and $U$. We explore a set of architectures in coarse-grained filtering and vary $(K, U)$ to measure the achieved AUC and the total training epochs in fine-grained refinement. Figure 12 shows that using a small $U$ (e.g., $U = 2$) can improve the final AUC while reducing total training epochs by enabling more candidates to be evaluated. We therefore set $U = 2$ in pTNAS.

We then examine the trade-off between $M$ and $K$ with varying $T_{\max}$. Figure 13 shows that a ratio $M/K \approx 30$ yields strong performance across different $T_{\max}$, which is adopted by our coordinator.

**Necessity of Coordinator**. To further illustrate the necessity and effectiveness of the coordinator, we fix the search time budget to 100 minutes and compare architectures obtained using different manually set $K$ values with those determined by the coordinator. Table 13 shows that the choice of $K$ heavily influences the search performance, and the coordinator-selected $K$ achieves the highest AUC. This confirms that the budget-aware coordinator is effective in holistically optimizing both coarse-grained filtering and fine-grained refinement phases and is necessary for achieving progressive NAS.

Table 13. Impact of $K$ on the searched AUC. Search for 100 mins.

|  | K=1 | K=10 | K=100 | K=500 | Coordinator (K=309) |
|---|---|---|---|---|---|
| AUC | 0.9789 | 0.9801 | 0.9802 | 0.9791 | **0.9805** |

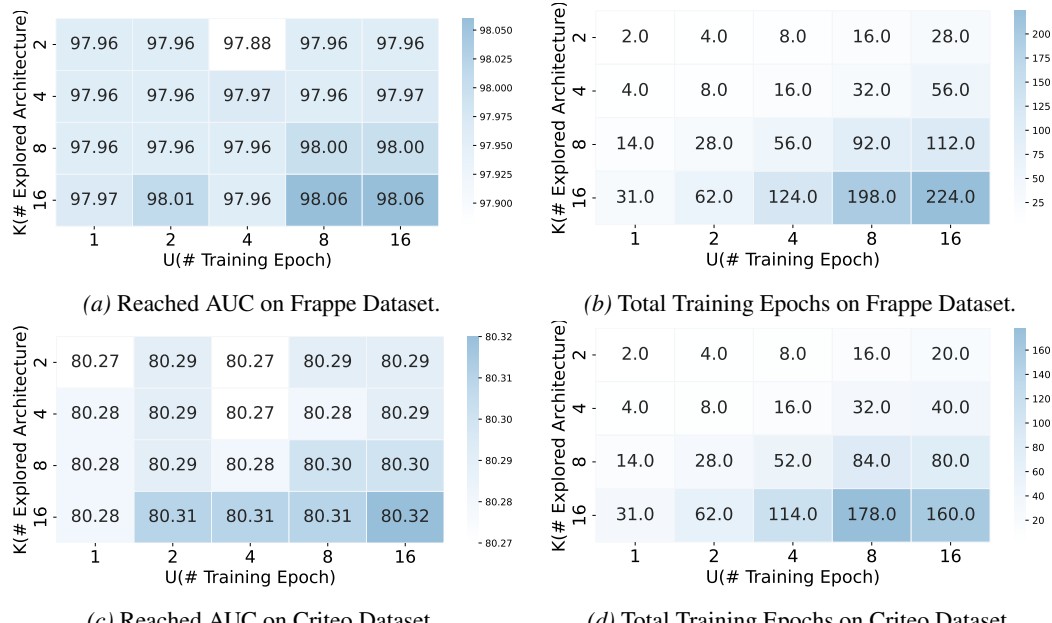

*(a)* Reached AUC on Frappe Dataset.

*(b)* Total Training Epochs on Frappe Dataset.

*(c)* Reached AUC on Criteo Dataset.

*(d)* Total Training Epochs on Criteo Dataset.

*Figure 12.* Trade-offs between $K$ and $U$ in the fine-grained refinement phase.

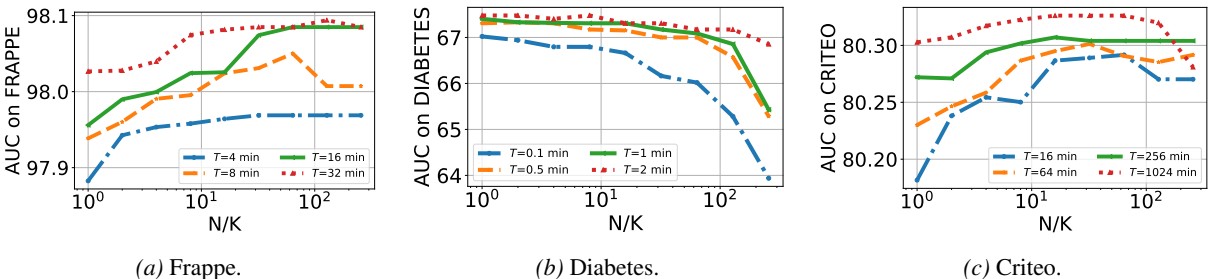

*(a)* Frappe.

*(b)* Diabetes.

*(c)* Criteo.

*Figure 13.* Trade-offs between $M$ and $K$ under different $T_{\max}$.

### F.4. Comparison and Analysis of zero-cost proxies

*Table 14.* Efficiency of zero-cost proxies measured on Frappe dataset (seconds). FC/BC: Forward/Backward computation, N/R: the target AUC is not reached. Search costs are evaluated on Frappe.

| | Grad Norm | NAS WOT | NTK Cond | NTK Trace | NTK TrAppx | Fisher | GraSP | SNIP | SynFlow | **pTProxy** |
|---|---|---|---|---|---|---|---|---|---|---|
| Time Complexity | 1FC 1BC | 1FC | 1FC 1BC | 1FC 1BC | 1FC 1BC | 1FC 1BC | 1FC 1BC | 1FC 1BC | 1FC 1BC | 2FC 1BC |
| Computational Cost | $2.69\times 10^{-3}$ | $1.76\times 10^{-3}$ | $5.75\times 10^{-3}$ | $5.66\times 10^{-3}$ | $2.85\times 10^{-3}$ | $9.59\times 10^{-3}$ | $1.14\times 10^{-2}$ | $3.02\times 10^{-3}$ | $2.27\times 10^{-3}$ | $2.70\times 10^{-3}$ |
| Search Cost for AUC 0.9793 coarse-grained filtering Only | 19.9 | N/R | N/R | N/R | N/R | N/R | N/R | 18.8 | 22.1 | **2.8** |
| Search Cost for AUC 0.9798 coarse-grained filtering + fine-grained refinement | 695 | 1188 | N/R | 7568 | 4178 | 1233 | 1141 | 726 | 1184 | **62** |

We report both theoretical complexity and empirical scoring time per architecture on Frappe, as well as search cost to reach a target AUC when using EA guided by different zero-cost proxies. Table 14 shows that pTProxy is computationally efficient

and achieves the lowest search cost among the compared proxies.

### F.5. Comparison of pTNAS with Additional Baselines

*Table 15.* Target AUC = 0.9798. N/R: the target AUC is not reached.

| NAS Approaches | Search Cost for AUC 0.9798 (Sec) |
| --- | --- |
| Training-based (RS) | 21560 |
| Training-based (RL) | N/R |
| Training-based (EA) | 8462 |
| Training-free (SNIP) | N/R |
| Training-free (NASWOT) | N/R |
| Training-free (SynFlow) | N/R |
| Training-free (pTProxy) | N/R |
| Warmup (NASWOT) | 234 |
| Warmup (SNIP) | 464 |
| Warmup (SynFlow) | 289 |
| Warmup (pTProxy) | **227** |
| Move-Proposal (NASWOT) | 9503 |
| Move-Proposal (SNIP) | 8659 |
| Move-Proposal (SynFlow) | 9106 |
| Move-Proposal (pTProxy) | 6940 |
| coarse-grained filtering + fine-grained refinement (pTProxy +Full Training) | 329 |
| pTNAS | **62** |

#### F.5.1. COMPARISON WITH DIFFERENT COMBINATIONS OF TRAINING-FREE AND TRAINING-BASED METHODS

In this section, we compare pTNAS with representative NAS variants that combine training-free and training-based evaluations. We include two strategies from prior work (Abdelfattah et al., 2021): the fully decoupled *warmup* strategy and the coupled *move proposal* strategy. In contrast, pTNAS employs a decoupled two-phase design: the coarse-grained filtering phase is guided by EA using zero-cost proxies, and the fine-grained refinement phase uses training-based evaluation scheduled by successive halving; the two phases are holistically optimized by a budget-aware coordinator (Section 3.5) to support progressive NAS.

To empirically compare search efficiency, we evaluate the time required to reach a target AUC under different combinations. Specifically, we compare: training-based only (RS/RL/EA), training-free only (best-performing zero-cost proxies), and Warmup/Move-Proposal (both using EA as in pTNAS with best-performing zero-cost proxies).

The results in Table 15 show that: (1) training-based only and Move-Proposal are the most time-consuming approaches due to repeated costly training; (2) training-free only fails to reach the target AUC; (3) pTNAS reaches the target AUC with substantially reduced time, outperforming Warmup and Move-Proposal; (4) the two-phase scheme with the coordinator achieves better search performance than other combinations, confirming the necessity of combining training-free and training-based evaluation in a budget-aware manner.

#### F.5.2. COMPARISON WITH ONE-SHOT NAS METHODS

One-shot NAS methods reduce search cost by training a single supernet and evaluating subnets via weight sharing (Liu et al., 2019; Pham et al., 2018). Their effectiveness depends on the correlation between the inherited-weight performance estimates and true performance after full training.

Given our DNN-based search space, we train a supernet with four layers, each with a maximum width of 512. We evaluate subnet performance using inherited weights and compare it with full-training performance on Frappe. The resulting SRCC is only 0.12, which is substantially lower than pTProxy (SRCC 0.82), leading to inferior search performance under weight sharing.

## F.6. Visualization of Correlation for zero-cost proxies

For each zero-cost proxy, we randomly sample 4000 architectures from *NAS-Bench-Tabular* and compute both their validation AUC after training and the zero-cost proxy score at initialization. We visualize the score–AUC relationship in Figures 14, 15, 16, 17, and 18.

Overall, these zero-cost proxy scores exhibit a positive association with validation AUC across datasets. In particular, pTProxy (right three plots in Figure 18) shows a more consistent monotonic trend with AUC, supporting its effectiveness as a training-free proxy for tabular DNN architectures.

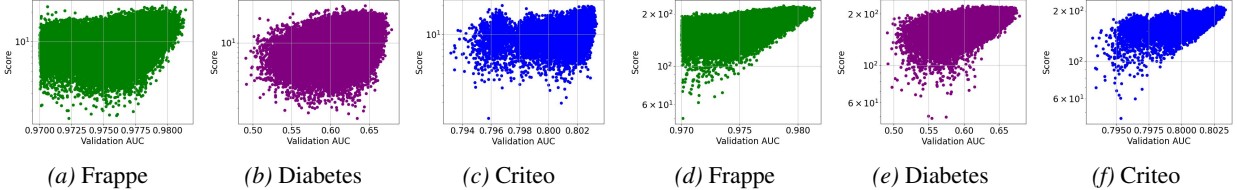

*Figure 14.* **GradNorm** (left three) and **NASWOT** (right three): proxy score vs. validation AUC.

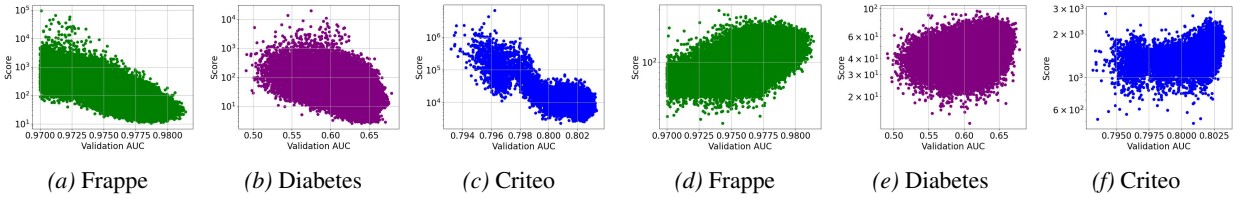

*Figure 15.* **NTKCond** (left three) and **NTKTrace** (right three): proxy score vs. validation AUC.

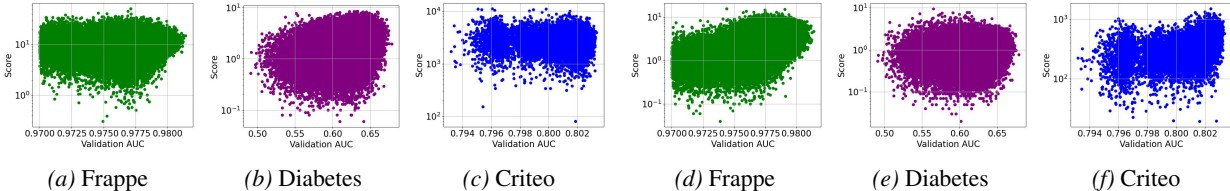

*Figure 16.* **NTKTraceAppx** (left three) and **Fisher** (right three): proxy score vs. validation AUC.

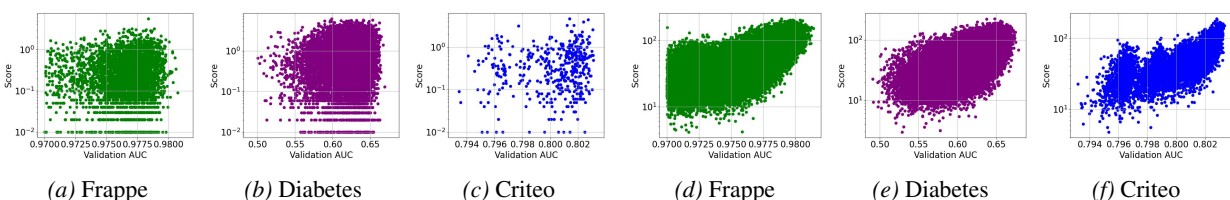

*Figure 17.* **GraSP** (left three) and **SNIP** (right three): proxy score vs. validation AUC.

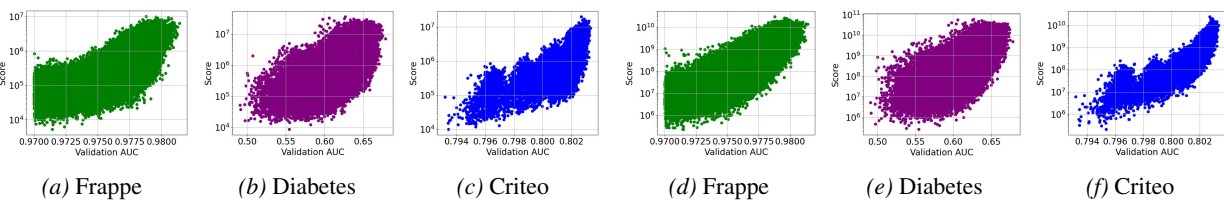

*Figure 18.* **SynFlow** (left three) and **pTProxy** (right three): proxy score vs. validation AUC.

*Table 16.* Effectiveness evaluation of pTNAS against existing tabular models.

| Task Type (Metric) | | Regression (MAE; ↓) | | | | | Classification (AUC; ↑) | | | | | Rank (↓) |
|---|---|---|---|---|---|---|---|---|---|---|---|---|
| Dataset/Task | | Event *user-attend* | Beer *beer-pos* | Trial *site-success* | HM *item-sales* | Rank *avg* | Event *user-repeat* | Beer *user-active* | Trial *study-out* | Avito *user-click* | Rank *avg* | Global Rank |
| Type | Method | | | | | | | | | | | |
| CM | LR | 0.3912 | 0.2046 | 0.4594 | 0.0659 | 12.25 | 0.7376 | 0.8812 | 0.6881 | 0.6407 | 11.25 | 11.75 |
| | RF | 0.3745 | 0.1997 | 0.4565 | 0.0584 | 10.63 | 0.7270 | 0.8737 | 0.6770 | 0.6450 | 12.75 | 11.69 |
| | CatBoost | 0.2607 | 0.1975 | **0.4429** | 0.0557 | 5.75 | 0.7429 | 0.9060 | 0.6945 | 0.6504 | 7.00 | 6.25 |
| | LightGBM | 0.2547 | 0.1903 | 0.4595 | 0.0495 | 5.63 | 0.7340 | 0.9061 | 0.6999 | 0.6527 | 7.00 | 6.31 |
| TFM | TabPFN | 0.3406 | 0.1903 | 0.4639 | 0.0583 | 9.88 | **0.7754** | 0.9164 | **0.7051** | 0.6437 | 5.00 | 7.44 |
| | TabICL* | - | - | - | - | - | 0.7692 | 0.9077 | 0.7018 | 0.6503 | **4.50** | **4.50** |
| DTM | DNN | 0.2523 | 0.1947 | 0.4500 | 0.0551 | 5.25 | 0.7294 | 0.9064 | 0.6830 | 0.6546 | 8.25 | 6.75 |
| | DeepFM | **0.2491** | 0.2091 | 0.4581 | 0.0541 | 7.50 | 0.7130 | 0.9047 | 0.7013 | 0.6283 | 10.50 | 9.00 |
| | FTTrans | 0.2539 | **0.1825** | 0.4290 | 0.0584 | **4.13** | 0.7346 | 0.9131 | 0.6836 | 0.6502 | 8.00 | 6.06 |
| | ARM-Net | 0.2642 | 0.1912 | 0.4468 | 0.0515 | 5.50 | 0.7402 | 0.9016 | 0.6965 | **0.6604** | 6.75 | 6.13 |
| LLM | TP-BERTa | 0.2768 | 0.3155 | 0.4612 | 0.3514 | 13.75 | 0.5457 | 0.5170 | - | 0.5122 | 16.00 | 14.71 |
| | Nomic | 0.2677 | 0.3439 | 0.4545 | 0.2063 | 12.25 | 0.6896 | 0.8896 | 0.6533 | 0.5771 | 13.75 | 13.00 |
| | BGE | 0.2645 | 0.2829 | 0.4511 | 0.0772 | 10.50 | 0.6787 | 0.8868 | 0.6503 | 0.6462 | 13.25 | 11.88 |
| NAS ($T_{max}$ = 10s) | TabNAS | 0.2635 | 0.1994 | 0.4520 | 0.0507 | 6.50 | 0.7512 | **0.9357** | 0.6902 | 0.6480 | 6.00 | 6.25 |
| | EA-NAS | 0.2639 | 0.2002 | 0.4485 | 0.0797 | 9.00 | 0.7518 | 0.9227 | 0.7017 | 0.6473 | 5.00 | 7.00 |
| | **pTNAS** | **0.2432** | 0.1794 | 0.4466 | **0.0497** | 1.75 | 0.7769 | **0.9370** | **0.7068** | **0.6680** | **1.00** | 1.38 |

∗ TabICL is inherently limited to classification objectives and is not applicable to regression-based tasks.

*Table 17.* **Average fitting, inference, and total time (seconds) per model, aggregated over all datasets in each task type**. Fitting includes NAS time if applicable: fit = NAS + train; Total = fit + inference.

Efficiency-effectiveness trade-off: while pTNAS incurs additional fitting overhead due to NAS, it delivers stronger predictive performance (lower MAE on regression and higher AUC on classification), achieving the best average ranks on both task types and the best global rank in Tab. 16 (Regression rank avg: 1.75; Classification rank avg: 1.00; Global rank: 1.38). Notably, although pTNAS is not always the fastest on classification, its inference latency remains small, and the accuracy gains are substantial.

| Type / Method | | Regression ↓ | | | Classification ↓ | | |
|---|---|---|---|---|---|---|---|
| | | Fitting (NAS+Train) | Inference | Total | Fitting (NAS+Train) | Inference | Total |
| CM | CatBoost | 804.49 | **0.22** | 804.71 | 47.24 | **0.04** | 47.28 |
| | LightGBM | **101.84** | **0.27** | 102.11 | **10.78** | 0.04 | **10.82** |
| TFM | TabPFN | 105.33 | 54.91 | 160.24 | 20.74 | 17.99 | 38.73 |
| DTM | DNN | 254.16 | 3.32 | 257.48 | 56.34 | 0.24 | 56.58 |
| | DeepFM | 149.47 | 0.35 | 149.82 | **9.66** | **0.12** | 9.78 |
| | FTTrans | 407.03 | 3.63 | 410.66 | 37.81 | 0.33 | 38.14 |
| NAS | **pTNAS** | 33.12 | 0.41 | 33.53 | 18.81 | **0.12** | 18.93 |

# G. Complete Results Across All Datasets

Table 16 reports the complete per-dataset results for the eight realistic multi-table datasets used in our paper. Table 17 reports the efficiency results in tabular form, corresponding to Figure 5.

