# OpenReview forum: "pTNAS: Progressive Neural Architecture Search for Tabular Data"
_ICML.cc/2026/Conference — ICML 2026 regular_

### Official Review · Reviewer_hKq6 · 2026-03-10

**Soundness:** 2
**Presentation:** 2
**Significance:** 2
**Originality:** 2
**Overall Recommendation:** 4
**Confidence:** 4

**Summary:**

The paper proposes a progressive neural architecture search (NAS) framework tailored for tabular data, which includes a coarse-grained training-free filtering phase and a training-based refinement phase. In the first stage, a novel zero-cost proxy called pTProxy is proposed. Experimental results show that the proposed method outperforms classical tabular models, tabular foundation models, and deep tabular architectures with high search efficiency.

**Compliance With Llm Reviewing Policy:**

Affirmed.

**Key Questions For Authors:**

1.	It is unclear how to mutate the architecture during the coarse-grained architecture filtering phase. The authors should introduce the mutation process in detail.
2.	How does the approach scale to large-scale datasets with high feature dimensionality?

**Limitations:**

yes

**Strengths And Weaknesses:**

**Strengths:**

1.	The paper designs a zero-cost proxy designed for tabular neural architectures, which helps to improve the search efficiency significantly.
2.	A benchmark containing 160k architectures is introduced for reproducibility and fair comparisons in tabular NAS research.
3.	Experimental results demonstrate the effectiveness and computational efficiency of the proposed method.

**Weaknesses:**

1.	The search space is restricted to MLP-style networks with variable depth and width. The author should explore more diverse search spaces, e.g., transformer-based architectures.
2.	The proposed NAS-Bench-Tabular appears to rely primarily on three core datasets, limiting the diversity of tabular domains.
3.	The design of zero-cost proxy is heuristic, lacking of theoretical justification. It is unclear whether the proxy generalizes to other architecture families or tasks. It is suggested to add theoretical analysis of pTProxy.
4.	Besides pTProxy, the overall novelty of the proposed method is limited. Both the evolutionary search in the first phase and the successive halving strategy in the second phase are very commonly-used methods in the AutoML field.

---

> ### Author Rebuttal · Authors · 2026-03-29
>
> We thank the reviewer for your valuable feedback. We address the concerns below, with supplementary results (**SR**) provided at this [link](https://anonymous.4open.science/r/pdf-5225/ICML.pdf).
> > W1: Limited search space.
>
> **Response:** pTProxy can also be extended to more complex spaces. Please refer to the **[Global Response] New Search Spaces** (under Reviewer Qefk) for details.
> > W2: Diversity of tabular domain.
>
> **[Global Response] Datasets**: We use 3 datasets to build NAS-Bench-Tabular for 2 reasons: (1) they are well-known tabular datasets covering various domains [1,2]; and (2) they already span large differences in scale (about 100K to 46M samples) and feature space (369 to 5,382 features), as summarized in Table 7.
> In addition, we evaluate the final search performance on 8 datasets from RelBench (Table 2), a large-scale benchmark over real relational databases [3].
>
> We further add two new datasets, Avito (ad-ctr) and HM (user-churn), to broaden the evaluation across task type, sample size, and feature dimensionality (Table 21 in [SR]).
> > W3: Theoretical analysis of pTProxy.
>
> **Response:** Our theoretical analysis of pTProxy is detailed in Appendix E. We show that the proposed neuron saliency can be written as an aggregation of synaptic saliency over both incoming and outgoing connections, linking pTProxy to trainability. Together with the layer-wise recalibration by width and sensitivity, this provides theoretical justification for why pTProxy captures both trainability and expressivity.
>
> Our evaluation on 3 search spaces, 13 datasets with different samples, features, task types, and domains, 2 NAS baselines, and 13 tabular model baselines shows that pTProxy generalizes well across diverse settings (see Sections 4.2 and 4.3 and Tables 17, 18, and 19 of [SR]).
> > W4: Novelty of the proposed method.
>
> **Response:** Compared with existing AutoML approaches: (1) *Benchmarking training-free proxies on tabular data*. We are the first to systematically benchmark training-free proxies on tabular DNNs, showing that many existing proxies have low correlation with true architecture performance and are inconsistent across datasets. (2) *A new training-free proxy tailored for tabular data*. We propose pTProxy, a tabular-specific proxy based on neuron saliency that captures both trainability and expressivity, and is empirically more effective and stable than existing proxies. (3) *Progressive NAS design for tabular data*. We formulate progressive NAS for tabular data, where the method should return a well-performing architecture even under small time budgets. To our knowledge, pTNAS is the first approach to explicitly support this through a two-phase design and a budget-aware coordinator. (4) *A tabular NAS benchmark*. We construct NAS-Bench-Tabular, the first tabular NAS benchmark with a DNN-based search space and fully trained statistics for 160K architectures on three representative tabular datasets with 20,622 GPU hours of training.
>
>
> > Q1: Architecture mutation.
>
> **Response:** (1) *NAS-Bench-Tabular* space: each mutation randomly selects one layer and replaces its width with another candidate value from the predefined width set (up to 20 choices). (2) *ResDNN* space: each mutation randomly performs one of three operations: replacing the width of one residual block with another candidate value, inserting a new block, or deleting an existing block, subject to the predefined depth range. (3) *BlockMixed* space: each mutation randomly inserts/deletes a block, or changes one block attribute, including block type, width, normalization, activation, dropout, connectivity, or the auxiliary parameter of attention/transformer blocks.
>
>
>
> > Q2: Scale to large-scale datasets with high feature dimensions?
>
> **Response:** pTNAS scales well to both high feature dimensions and large datasets: (1) *high feature dimension*: the cost of pTProxy is primarily determined by block width and depth, whereas the feature dimension affects only the first layer. As shown below, increasing the feature dimension from 10 to 5000 changes the runtime only marginally.
> | Model | feat\_dim=10 | feat\_dim=100 | feat\_dim=1000 | feat\_dim=5000 |
> | - | -: | -: | -: | -: |
> | ResDNN 4-block | 2.4ms | 2.4ms | 2.5ms | 2.5ms |
> | BlockMixed 3-block | 1.7ms | 1.7ms | 1.8ms | 1.8ms |
>
> (2) *Large datasets*: pTProxy is computed on a small mini-batch, so the filtering phase remains efficient and quickly removes unpromising architectures. The refinement phase trains only the top-$K$ candidates with successive halving. This is much more scalable than training many architectures from scratch, and is consistent with our results on large-scale datasets such as Criteo.
>
> [1] ARM-Net: Adaptive Relation Modeling Network for Structured Data [SIGMOD 2021]
>
> [2] TabNAS: Rejection Sampling for Neural Architecture Search on Tabular Datasets [NeurIPS 2022]
>
> [3] RELBENCH: A Benchmark for Deep Learning on Relational Databases [NeurIPS 2024]

---

> > ### Author Rebuttal · Reviewer_hKq6 · 2026-04-01
> >
> > Thanks for your rebuttal. I have increased my score accordingly.

---

> > > ### Author Response · Authors · 2026-04-01
> > >
> > > We are glad to hear that your concerns have been fully resolved! We sincerely thank you for your thoughtful feedback and for increasing the score. We will incorporate the additional results and discussions in the revised manuscript accordingly.

---

### Official Review · Reviewer_Qefk · 2026-03-10

**Soundness:** 3
**Presentation:** 3
**Significance:** 2
**Originality:** 2
**Overall Recommendation:** 5
**Confidence:** 3

**Summary:**

This paper presents pTNAS, a progressive neural architecture search approach for tabular data. pTNAS first uses zero-cost proxy scores and a regularized evolutionary algorithm to propose a set of candidate architectures. It then applies successive halving (SH) to optimize for the optimal architecture for the given task. Experiments show that pTNAS achieves a better overall performance and lower inference latency on the tabular benchmark tasks.

**Compliance With Llm Reviewing Policy:**

Affirmed.

**Final Justification:**

The rebuttal addresses my main concern about the paper. So I increased my score.

**Key Questions For Authors:**

Many details are still missing for the experimental setup:

* In Figure 3, it takes at least 100 minutes for pTNAS to achieve the near-optimal AUC score. Why does it only take 7s for pTNAS to search for the optimal architecture? How much time do $t_1$ and $t_2$ in Equation (6) cost exactly?
* How do the authors determine the parameter $\eta$ in Equatino (6)?
* If I understand Equation (4) correctly, computing $d_l$ requires at least two forward passes (one with x and another one with the perturbation ). Why does the complexity in Table 4 only require 1 forward pass and 1 backward pass?
* Are all the baseline models' hyperparameters also tuned (with the same budget provided to pTNAS)

**Limitations:**

My main concern for this paper is the limited hyperparameter search space. It would be interesting to see how the approach works with a larger search space and more diverse architectures.

**Strengths And Weaknesses:**

**Strength**

Overall, the idea of progressively decreasing the candidate pool and the SH-based budget-aware refinement phase is interesting.  The experiments also show a promising result.

**Weakness**

* The hyperparameter search space in this work is still limited. All networks in the search space contain only a 4-layer (line 1019) dense feedforward network with fixed architecture; the only difference among these networks is network width. However, the network depth might also influence pTProxy's performance, given the discussion for equation 4. The correlation between network depth and pTProxy score has not been clearly studied. Is there any reason for only searching for an architecture with a fixed network depth? How would the zero-cost proxy and final architecture perform if the number of layers being part of the hyperparameter search space? Additionally, other hyperparameters, such as the dropout rate, normalizer layer types, and activation functions, are also not discussed.

**small remarks**
* Figure 5 is a bit confusing; the scale for inference time does not seem to match the scales for training time. For instance, Table PDF takes 105 seconds to fit and 54.9 seconds to infer; however, the purple bar is much larger than the yellow bar.

---

> ### Author Rebuttal · Authors · 2026-03-29
>
> We thank the reviewer for your valuable feedback. We address the concerns below, with supplementary results (**SR**) provided at this [link](https://anonymous.4open.science/r/pdf-5225/ICML.pdf).
> > W1/L1: Limitations of the search space.
>
> **[Global Response] NAS-Bench-Tabular Motivation**: Existing studies show that DNNs (e.g., MLPs) with fixed or modest depth and properly tuned hidden sizes could already achieve top performance on tabular data [3,4,5], and recent work [1,2] further shows that even tabular foundation models can be distilled into compact MLPs that preserve most of the accuracy while achieving much lower latency. Motivated by this, we adopt a fixed-depth DNN search space and focus on exploring layer-wise configurations. To build NAS-Bench-Tabular, we fully train and evaluate 160K architectures, which costs around 20,622 GPU hours.
>
> **[Global Response] New Search Spaces**: We add 2 search spaces: (1) **ResDNN**, a DNN-based space (2,500 architectures) with variable depth, variable width, and residual (skip) connections; and (2) **BlockMixed**, a more heterogeneous space (1,000 architectures) composed of MLP, attention, transformer, and residual (skip) blocks, each with its own width, as shown in Figure 17(a) of [SR]. BlockMixed also varies block-level normalization (LayerNorm / BatchNorm), activation (ReLU / GELU / SiLU), and dropout rate (0.0 / 0.1 / 0.2).
>
> For evaluation, we use two representative datasets, Event user-attendance (small-scale, high-dimensional, regression) and Avito user-clicks (large-scale, low-dimensional, classification), together with two new datasets, Avito (ad-ctr) and HM (user-churn). Table 17 of [SR] shows that pTProxy consistently achieves the highest SRCC, and Table 18 shows that pTNAS achieves stronger search performance under practical budget constraints.
>
> In conclusion, the consistent improvements across three search spaces of increasing complexity indicate that pTNAS is not tied to any specific architecture design and generalizes well across diverse settings.
>
> > Remark: Figure 5 is a bit confusing.
>
> **Response:** We apologize for the confusion. This is a plotting error, and we have corrected it.
> > Q1: Time usage in Equation (6).
>
> **Response:** Figure 3 and the 7s setting correspond to different regimes. Figure 3 studies pTNAS under much larger budgets (up to 100 mins) to show progressive search toward near-optimal AUC, whereas the 7s is a deliberately constrained budget chosen to match the tuning cost of LightGBM for a fair comparison.
>
> Under 7s, pTNAS can already outperform strong baselines (TabPFN) while maintaining efficiency comparable to LightGBM, which is the main objective in Section 1. For example, on Event (user-attendance) dataset with $T_{\max}=7$s, the coordinator spends less than 1s determining the plan, allocates $M=90$ for filtering ($T_1=1.4$s), and keeps $K=3$ architectures for refinement with $U=2$ epochs ($T_2=5.4$s). With larger budgets, pTNAS can continue searching and discover better architectures; see Table 18 of [SR].
> > Q2: Determine $\eta$ in Equation (6)?
>
> **Response:** $\eta$ controls how aggressively architectures are pruned in each round. We use the default setting of $\eta=3$, as in the original Hyperband paper [6], which empirically performs well across the experiments.
> > Q3: Number of forward passes.
>
> **Response:** We implement pTProxy with one batched forward and one backward: the forward jointly evaluates x and its perturbed version $x+ \epsilon \delta$, while the backward is applied only to the original x's outputs. The perturbed branch is used only to estimate $d_l$ in Eq. (4). Adding the perturbation branch incurs only millisecond-level overhead per architecture: 0.40 ms on Frappe, 2.65 ms on UCI Diabetes, and 1.44 ms on Criteo.
> > Q4: Baseline models hyperparameter tuning.
>
> **Response:** Yes. For the classical models, we tune the model hyperparameters for each dataset using the built-in tuning methods in torch_frame [7]. For LightGBM, this tuning process takes a median of about 7 seconds across datasets. For fair comparison, we hence use the same tuning budget for both the deep tabular models and our pTNAS. If a deep tabular model cannot complete a single tuning trial within this budget, we fall back to its default architecture.
>
>
> [1] TabPFN-2.5: Advancing the State of the Art in Tabular Foundation Models [arXiv 2026]
>
> [2] MotherNet: Fast Training and Inference via Hyper-Network Transformers [ICLR 2025]
>
> [3] Better by default: Strong pre-tuned MLPs and boosted trees on tabular data [NeurIPS 2024]
>
> [4] TabNAS: Rejection Sampling for Neural Architecture Search on Tabular Datasets [NeurIPS 2022]
>
> [5] Well-tuned simple nets excel on tabular datasets [NeurIPS 2021]
>
> [6] Hyperband: A novel bandit-based approach to hyperparameter optimization. [JMLR 2017]
>
> [7] https://pytorch-frame.readthedocs.io/en/latest/generated/torch_frame.gbdt.GBDT.html?highlight=tune#torch_frame.gbdt.GBDT.tune

---

> > ### Author Rebuttal · Reviewer_Qefk · 2026-04-01
> >
> > Thank you for your detailed response. That solves all my concerns. I will increase my score.

---

> > > ### Author Response · Authors · 2026-04-01
> > >
> > > We are very glad to know that our response has adequately addressed your concerns. We truly appreciate your time and your decision to raise your score for our paper. Following your suggestion, we will revise the main manuscript to expand the search space, clarify the experimental setup, and improve the presentation.

---

### Official Review · Reviewer_WCHE · 2026-03-11

**Soundness:** 3
**Presentation:** 3
**Significance:** 2
**Originality:** 2
**Overall Recommendation:** 4
**Confidence:** 3

**Summary:**

This paper introduces pTNAS, a  Neural Architecture Search (NAS) framework specifically designed for tabular data. The authors identify a key gap: existing NAS methods for tabular data are computationally expensive, while efficient methods from computer vision don't translate well due to the heterogeneous, non-structural nature of tables.
To address this, pTNAS employs a **filter-and-refine optimization strategy** coordinated by a budget-aware mechanism.
The paper makes a contribution by introducing **NAS-Bench-Tabular**, the first benchmark dataset for tabular NAS, containing over 160K unique architectures evaluated on multiple datasets. This facilitates reproducible research in the field.

Extensive experiments show that pTNAS achieves good performance, outperforming classical models (GBDTs), deep tabular models, and even tabular foundation models (like TabPFN) on a range of classification and regression tasks. It also demonstrates massive speedups (up to 82.75×) in search efficiency compared to traditional training-based NAS methods.

**Compliance With Llm Reviewing Policy:**

Affirmed.

**Final Justification:**

The author made a detailed rebuttal, and the reply basically solved my question, so I decided to improve my score.

**Key Questions For Authors:**

1.  Your search space is constrained to tuning the widths of a 4-layer MLP. How do you envision pTNAS scaling or adapting to more complex search spaces that include variable depth, different activation functions, or modern architectural components like feature transformers or attention mechanisms?
2.  The coordinator's optimal settings (e.g., M/K ratio ≈ 30, U=2) were derived from experiments on the NAS-Bench-Tabular datasets. How robust are these settings? If pTNAS were applied to a dataset with a very small number of samples or an extremely high feature dimensionality, would the coordinator need to be re-calibrated, or can it dynamically find new optimal trade-offs on the fly?
3.  pTProxy demonstrably outperforms other zero-cost proxies. Could you provide more intuition, beyond the mathematical derivation, for *why* aggregating SynFlow-style saliency at the neuron level (and taking the absolute value) is particularly well-suited for capturing complex feature interactions in tabular data?
4.  The progressive search curves (Figure 3) show pTNAS reaching a performance plateau relatively quickly. Is this plateau the true global optimum of the search space, or is it a limitation imposed by the proxy's estimation power or the refinement phase's early stopping? How could the framework be adjusted to continue making meaningful progress on much longer time horizons (e.g., days)?

**Limitations:**

yes

**Strengths And Weaknesses:**

Strengths:
1.  The paper clearly identifies a real problem—the inefficiency of tabular NAS—and proposes a logical, two-phase solution that combines the speed of zero-cost proxies with the accuracy of full training.
2. This is a major contribution to the community. It provides a standardized platform for future research, enabling fair and reproducible comparisons of tabular NAS algorithms, which was previously a significant hurdle.
3. The design and theoretical justification of pTProxy, which integrates both trainability and expressivity, is a strong contribution. The empirical results in Table 1 demonstrate its clear superiority over adapted vision-based proxies.
4.   The paper includes a wide range of experiments, from benchmarking the proxy itself to comparing the full pTNAS pipeline against numerous strong baselines (CMs, DTMs, TFMs, LLMs) across multiple datasets and task types. The analysis of progressive performance (Figure 3) is particularly compelling.

Weaknesses:
1.  The NAS-Bench-Tabular and the search space used for pTNAS only consider varying the *widths* of a fixed-depth MLP. While effective, this omits other important architectural dimensions like depth, activation functions, connectivity patterns (e.g., skip connections), and different layer types. This limits the generality of the findings.
2. The performance of pTProxy, like all zero-cost proxies, is contingent on the network's initialization scheme (Xavier, He, etc.). While the paper ablates this, the sensitivity to initialization could be a point of instability when applying the method to entirely new, unseen architectures or datasets not covered in the benchmark.
3. While the final formulation is clean, the theoretical derivation linking neuron saliency to SynFlow (in Appendix E) is quite mathematically involved. This might make it difficult for practitioners to fully grasp *why* it works so well, potentially hindering adoption or further innovation.
4.  The coordinator's hyperparameters (like the optimal M/K ratio of ~30 and U=2) are empirically determined on the benchmark datasets. The paper doesn't deeply explore how sensitive pTNAS is to these values on drastically different datasets (e.g., with far fewer samples or a different number of features) and whether the coordinator's internal model can dynamically adjust them without relying on these pre-set "good" values.

---

> ### Author Rebuttal · Authors · 2026-03-29
>
> We thank the reviewer for your valuable feedback. We address the concerns below, with supplementary results (**SR**) provided at this [link](https://anonymous.4open.science/r/pdf-5225/ICML.pdf).
> > W1: Search space limitations.
>
> > Q1: Adapting pTNAS to more complex search spaces.
>
> **Response:** We have added 2 new search spaces. Please refer to the **[Global Response] New Search Spaces** (under Reviewer Qefk) for details.
>
> For the motivation for using a DNN-based search space, please refer to **[Global Response] NAS-Bench-Tabular Motivation** (under Reviewer Qefk) for details.
> > W2: Sensitivity to initialization.
>
> **Response:** We add experiments to evaluate pTProxy’s SRCC under Xavier, He, and LeCun on 2 new search spaces across 4 datasets. As shown in Table 20 of [SR], the SRCC remains stable, indicating that different initializations may change absolute scores but have a limited effect on the ranking.
>
> We further evaluate the final pTNAS performance on BlockMixed and with HM (user-churn). Different methods produce slightly different architectures but nearly identical final test AUCs, suggesting that both pTProxy and pTNAS are reasonably stable to initialization.
> | Init Method | Searched Architecture| Test AUC |
> | - | - | - |
> | Xavier      | mlp-24, trans-64, mlp-16, skip-64 | 0.6317   |
> | He          | mlp-16, trans-64, mlp-16, skip-64 | 0.6317   |
> | LeCun       | mlp-24, trans-32, mlp-24, skip-64 | 0.6316   |
>
> > W3: Theoretical derivation.
>
> > Q3: Intuition of pTproxy.
>
> **Response:** Our intuition is that strong tabular performance often depends on cross-feature interactions [1,2,3], which are naturally expressed by hidden neurons that combine signals from multiple input features and act as local nonlinear detectors of useful feature patterns. In pTProxy, we therefore aggregate SynFlow-style saliency at the neuron level to better reflect whether an architecture can form such detectors [4,5]. Further, the use of absolute value computation prevents mutual cancellation of neuron saliency, which ensures the importance of all neurons is captured. All these designs allow pTProxy to better capture both expressivity and trainability of an architecture.
> > W4/Q2: Robustness of the coordinator's hyperparameters.
>
> **Response:** $M$ controls exploration, $K$ controls refinement, and $U$ is the unit budget in successive halving. In our sensitivity analysis in Appendix F.3, we tune these settings on one dataset and transfer them to the other two, finding that $U=2$ and $M/K \approx 30$ *work well across datasets and budgets*. We use the same settings on the RelBench tasks (Table 2) and still obtain strong results.
>
> We further test the same coordinator settings on 2 newly added datasets, Avito (ad-ctr), a small-sample dataset with only 5.1K training samples, and HM (user-churn), across all 3 search spaces (NAS-Bench-Tabular, BlockMixed, and ResDNN). First, we measure the correlation between the 2-epoch and final performance and find that it remains positive and reasonably strong (see the table below). Second, we evaluate the final searched performance across the 3 search spaces; Tables 17, 18, and 19 of [SR] show that pTNAS still achieves strong results.
> | Search Space      | SRCC  |
> | - | - |
> | BlockMixed        | 0.639 |
> | NAS-Bench-Tabular | 0.626 |
>
> Overall, across 13 datasets and 3 search spaces, the coordinator settings appear reasonably robust. Since relatively small and simple architectures can already achieve top performance on tabular data, $U=2$ is often sufficient to eliminate weak candidates early, while $M/K \approx 30$ provides a practical balance between exploration and refinement.
> > Q4: Progressive search curves with longer time.
>
> **Response:** Yes. In Figure 3, the red line is the true global optimum of the search space, obtained by fully training and evaluating 160K architectures with about 20,622 GPU hours. pTNAS closely approaches this optimum quickly because it filters weak candidates very quickly, while the refinement phase focuses training only on promising ones.
>
> With more budget, pTNAS can simply continue from the current search state: it skips architectures that have already been explored, keeps the best explored ones as the current population, and continues exploring new candidates and refining the most promising ones.
>
> [1] AutoInt: Automatic Feature Interaction Learning via Self-Attentive Neural Networks [CIKM 2019]
>
> [2] TabNet: Attentive Interpretable Tabular Learning [AAAI 2021]
>
> [3] FIVES: Feature Interaction via Edge Search for Large-Scale Tabular Data [KDD 2021]
>
> [4] Pruning Neural Networks without Any Data by Iteratively Conserving Synaptic Flow [NeurIPS 2020]
>
> [5] Importance Estimation for Neural Network Pruning [CVPR 2019]

---

> > ### Author Rebuttal · Reviewer_WCHE · 2026-04-02
> >
> > Thank you for your reply. I raise my score.

---

> > > ### Author Response · Authors · 2026-04-02
> > >
> > > We are very glad to hear that your concerns have been fully resolved! We truly appreciate your valuable comments and your decision to raise the score. We will incorporate the additional results and discussions into the paper accordingly.

---

### Official Review · Reviewer_AFCC · 2026-03-19

**Soundness:** 2
**Presentation:** 2
**Significance:** 3
**Originality:** 3
**Overall Recommendation:** 5
**Confidence:** 3

**Summary:**

This work studies NAS for tabular data. The authors propose a novel zero cost proxy and progressive NAS search method that combines training-free and training-based evaluation. Further, they introduce the first tabular NAS benchmark and show a strong anytime performance of their method compared to previous NAS methods and strong performance against manually designed models.

**Compliance With Llm Reviewing Policy:**

Affirmed.

**Key Questions For Authors:**

* "We propose pTNAS, the first NAS approach supporting progressive NAS on tabular data" Why are other progressive NAS approaches not applicable to support tabular data?
* Which parts of your methodology are tailored / unique to tabular data?
* Do RE-NAS and TabNAS use the same architecture search space and training pipeline as you do?
* Where do the performance improvements of pTNAS stem from compared to existing approaches? Which parts of your method contribute how?

**Limitations:**

* NAS methods only evaluated on 3 datasets and 1 architecture search space, so generalization is unclear

**Strengths And Weaknesses:**

Strengths:
+ How to search for neural architectures for the tabular data modality is an underexplored and important problem setting.
+ The authors introduce the first NAS Benchmark for the tabular settings that will facilitate future research on this topic and broadens the evaluation of NAS algorithms in general
+ The document is well structured and easy to follow
+ Combining training-free and training-based architecture evaluation is an interesting idea that empirically finds strong architectures quicker than baselines (although the final performance is equal)
+ Their found architectures perform strongly against manually designed tabular models

Weaknesses:
- While the paper covers all the necessary background, a dedicated discussion on how this work differs to related work and where it's limitations lie is missing. Citations at key places are missing, e.g., "these datasets are commonly used in prior tabular NAS studies" (line 132), which prior tabular NAS studies and where are these discussed in the paper?
- It is unclear to me how the proposed zero cost proxy is specialized for tabular data.
- Evaluation is limited to only very few datasets and only one architecture search space that were likely also used in the development of the method. This is a general problem of this literature, which makes it questionable how well the zero cost proxy and search method would generalize
- In the comparison to manually designed methods, the performance of other NAS approaches is not shown

Minor point
* "HyperBand (Falkner et al., 2018), and etc." you wanted to continue writing that sentence?

---

> ### Author Rebuttal · Authors · 2026-03-29
>
> We thank the reviewer for your valuable feedback. We address the concerns below, with supplementary results (**SR**) provided at this [link](https://anonymous.4open.science/r/pdf-5225/ICML.pdf).
> > W1: Related work, limitations, and dataset citations.
>
> **Response:** **Related work**: pTNAS advances existing works in 3 fundamental aspects:
> (1) *Data-Adaptive Architectures*: Unlike traditional tabular models (see Table 2) that are constrained to static architectures, pTNAS adaptively searches for optimal DNN given the task. (2) *Training-free Search*: Different from existing tabular NAS methods like TabNAS [1] that rely on expensive training-based search, pTNAS is the first to introduce a training-free paradigm for efficient NAS on tabular data. (3) *Tabular-specific Zero-Cost Proxy (ZCP)*: Existing ZCPs are primarily designed for vision tasks, while pTProxy is the first proxy tailored specifically for tabular data, which explicitly characterizes key properties central to tabular deep learning, e.g., expressivity in terms of high-order feature interactions. We have expanded the related work section accordingly.
>
> **Limitation**: pTNAS optimizes search efficiency and predictive performance, but does not explicitly consider deployment constraints, e.g., memory footprint.
>
> **Dataset Citations**: Datasets such as Criteo are commonly benchmarked in prior tabular NAS works, e.g., TabNAS [1]. We have updated the citations and expanded the details in Appendix C.1.
> > W2/Q2: Specialization for tabular data.
>
> **Response:** We clarify the tabular-specific design of our methodology below: (1) *Tabular-specific ZCP*: Unlike vision tasks that rely on spatial priors (e.g., translation invariance), tabular learning is characterized by the need to capture *implicit and high-order feature interactions* among heterogeneous features [3]. As neurons are the basic feature-extraction units for modeling interactions in tabular DNNs, our pTProxy operates at the **neuron level** (as compared to the parameter level in existing ZCPs), which combines neuron activations (feature signals) with their gradients (task importance) to measure architecture quality. (2) *Application-driven Approach*: Tabular domains like recommendation and e-commerce often require identifying a viable model quickly and refining it as budget allows. Our pTNAS is the first tabular NAS approach to support this **progressive anytime capability** through a coordinated two-phase optimization scheme that balances fast training-free filtering with effective training-based refinement.
> > W3/L1: Limited datasets and search space.
>
> **Response:** For search space, please refer to **[Global Response] NAS-Bench-Tabular Motivation** (under Reviewer Qefk). We use 3 datasets for NAS benchmarking and 8 for final model comparison against the 13 baselines (Table 2). Please refer to **[Global Response] Datasets** (under Reviewer hKq6). For the generalization analysis, please refer to **[Global Response] New Search Spaces** (under Reviewer Qefk).
> > W4: Lack of NAS performance.
>
> > Q3: Search space and training of RE-NAS and TabNAS.
>
> **Response:** Yes. RE-NAS and TabNAS use the same search space and training pipeline as pTNAS.
> As suggested, we add experiments of them on 10 datasets (Table 19 of [SR]). As both are training-based approaches, they explore far fewer architectures than pTNAS under the same budget and thus perform worse.
> > M: "HyperBand sentence?
>
> **Response:** We have removed "and etc." to make the description clearer.
> > Q1: Other progressive NAS on tabular data.
>
> **Response:** Existing progressive NAS (e.g. PNAS[2]) are not directly applicable to tabular data for 3 reasons: (1) *Search space*: they are designed for vision-specific CNN cell-based search spaces rather than tabular architectures; (2) *Evaluation cost*: they are still training-based and thus are inefficient for tabular NAS under tight budgets; and (3) *Surrogate design*: their predictors rely on architecture encodings instead of tabular-specific, data-aware signals, and thus does not explicitly capture properties central to tabular learning.
> > Q4: Performance improvements of pTNAS.
>
> **Response:** (1) *Against NAS methods*: the improvement comes from the filter-and-refine design, which uses pTProxy for fast filtering and successive halving for accurate refinement. (2) *Against baseline models (Table 2)*: pTNAS performs better since it searches for a dataset-specific DNN instead of using a fixed backbone globally. (3) *Component contribution*: without pTProxy, exploration becomes much less efficient; without refinement, final selection becomes less reliable. Table 18 of [SR] confirms that full pTNAS consistently outperforms a no-refinement variant across all budgets.
>
> [1] TabNAS: Rejection Sampling for Neural Architecture Search on Tabular Datasets [NeurIPS 2022]
>
> [2] Progressive Neural Architecture Search [ECCV 2018]
>
> [3] AutoInt: Automatic Feature Interaction Learning via Self-Attentive Neural Networks [CIKM 2019]

---

> > ### Author Rebuttal · Reviewer_AFCC · 2026-03-31
> >
> > All my concerns have been adequately addressed and I am raising my score. I have not read the other reviews and the rebuttals.

---

> > > ### Author Response · Authors · 2026-03-31
> > >
> > > **Thank you for raising your score!**
> > >
> > > We are very glad to know that our response has adequately addressed your concerns. We would also like to thank you for raising your score for our paper. As per your suggestion, we will update the main paper to include two new search spaces, along with additional evaluations on new datasets. We will also further improve the clarity of the presentation to avoid potential confusion. Thank you again for your constructive comments.

---

### Decision · Program_Chairs · 2026-04-30

**Decision:**

Accept (regular)

**Comment:**

The paper proposes a new neural architecture strategy for designing better models for tabular data. Reviewers initially raised concerns about the empirical evaluation, but these were addressed during the rebuttal phase. Following the rebuttal, all reviewers voted to accept the paper.